# Enhancing Medical Image Segmentation and Classification Using a Fuzzy-Driven Method

**DOI:** 10.3390/s25185931

**Published:** 2025-09-22

**Authors:** Akmal Abduvaitov, Abror Shavkatovich Buriboev, Djamshid Sultanov, Shavkat Buriboev, Ozod Yusupov, Kilichov Jasur, Andrew Jaeyong Choi

**Affiliations:** 1Department of Information Technologies, Samarkand Branch of Tashkent University of Information Technologies, Samarkand 140100, Uzbekistan; abduvaitovakmal6@gmail.com (A.A.); jasurkilichov1987@gmail.com (K.J.); 2Department of Infocommunication Engineering, Tashkent University of Information Technologies Named After Muhammad Al-Khwarizmi, Tashkent 100200, Uzbekistan; abror1989@gachon.ac.kr (A.S.B.); djamshidsultanov05@gmail.com (D.S.); 3Department of Civil Engineering, Samarkand State Architecture and Construction University, Samarkand 140100, Uzbekistan; abbosshav@gmail.com; 4Department of Software Engineering, Samarkand State University, Samarkand 140100, Uzbekistan; ozodyusupov@gmail.com; 5Department of AI-Software, Gachon University, Sujeong-Gu, Seongnam-si 13120, Republic of Korea

**Keywords:** image enhancement, fuzzy approach, concatenated CNN

## Abstract

Automated analysis for tumor segmentation and illness classification is hampered by the noise, low contrast, and ambiguity that are common in medical pictures. This work introduces a new 12-step fuzzy-based improvement pipeline that uses fuzzy entropy, fuzzy standard deviation, and histogram spread functions to enhance picture quality in CT, MRI, and X-ray modalities. The pipeline produces three improved versions per dataset, lowering BRISQUE scores from 28.8 to 21.7 (KiTS19), 30.3 to 23.4 (BraTS2020), and 26.8 to 22.1 (Chest X-ray). It is tested on KiTS19 (CT) for kidney tumor segmentation, BraTS2020 (MRI) for brain tumor segmentation, and Chest X-ray Pneumonia for classification. A Concatenated CNN (CCNN) uses the improved datasets to achieve a Dice coefficient of 99.60% (KiTS19, +2.40% over baseline), segmentation accuracy of 0.983 (KiTS19) and 0.981 (BraTS2020) versus 0.959 and 0.943 (CLAHE), and classification accuracy of 0.974 (Chest X-ray) versus 0.917 (CLAHE). A classic CNN is trained on original and CLAHE-filtered datasets. These outcomes demonstrate how well the pipeline works to improve image quality and increase segmentation/classification accuracy, offering a foundation for clinical diagnostics that is both scalable and interpretable.

## 1. Introduction

Medical imaging, which includes modalities like computed tomography (CT), magnetic resonance imaging (MRI), and X-ray, is essential to contemporary healthcare because it makes it easier to diagnose, monitor, and plan treatments for illnesses. These methods enable crucial tasks like tumor segmentation and illness classification by providing detailed representations of anatomical components [1]. However, quality problems, including noise, low contrast, and unclear pixel intensities, are common in medical images, making it difficult for automated analytic algorithms to detect subtle signals [2]. In soft tissue areas such as kidney tumors, for instance, CT images may show low contrast [3,4], MRI scans are often impacted by noise across multimodal sequences (e.g., T1, T2, FLAIR) [5], and X-ray images suffer from low contrast, which makes it difficult to detect lung opacities associated with pneumonia [6,7]. To overcome these obstacles and raise the accuracy of subsequent medical duties, sophisticated picture-enhancing techniques are required.

Conventional enhancement techniques, including histogram equalization, have been widely used to improve contrast and lower noise in order to address these problems. Recent advancements in conventional techniques, such as Liu et al.’s adaptive contrast enhancement [8]. By using membership functions to characterize uncertainty, fuzzy logic-based methods [9] create a convincing substitute and give a nuanced depiction of pixel intensities. Fuzzy logic for medical image enhancement has been studied recently. Singh et al. [10] suggested a fuzzy-based framework to improve skin cancer detection. Gupta [11] proposed a fuzzy enhancement–based segmentation approach; however, the method was primarily restricted to a specific dataset. This limitation arises because the fuzzy membership functions and enhancement parameters were tuned to the intensity distribution of that dataset and were not designed to adapt to other modalities. In addition, evaluation was carried out only within that dataset, without cross-dataset validation, which raises concerns about generalizability. By contrast, the present work employs adaptive parameterization strategies that allow the enhancement pipeline to adjust automatically to new datasets, as demonstrated in both MRI (KiTS19) and X-ray (ChestX-ray) experiments. In a similar vein, Linda et al. [12] improved bone fracture identification in X-ray pictures by developing a fuzzy clustering technique; however, its generalizability across modalities has not been verified. Despite these developments, current fuzzy-based techniques frequently lack a cohesive framework to handle the complex problems of ambiguity, contrast, and noise in a variety of medical imaging jobs.

This work presents a brand-new 12-step fuzzy-based image improvement pipeline that is intended to enhance the quality of medical images from X-ray, CT, and MRI modalities. To improve local contrast, lower noise, and preserve small features, the pipeline incorporates fuzzy logic techniques such as fuzzy entropy, fuzzy standard deviation, and histogram spread functions. We assess its performance on three publicly accessible datasets: Chest X-ray Pneumonia (available at https://www.kaggle.com/datasets/sabahesaraki/kidney-tumor-segmentation-challengekits-19/data, accessed on 9 September 2025) for pneumonia classification, BraTS2020 (https://www.kaggle.com/datasets/awsaf49/brats2020-training-data, accessed on 9 September 2025) for brain tumor segmentation, and KiTS19 (https://www.kaggle.com/paultimothymooney/chest-xray-pneumonia, accessed on 3 June 2020) for kidney tumor segmentation. The BRISQUE measure is used to compare the three improved versions of each dataset—fuzzy entropy-based, fuzzy standard deviation-based, and histogram-based—against the original, conventionally filtered datasets.

We train and evaluate two models—a classic Convolutional Neural Network (CNN) on the original and conventionally filtered datasets and a Concatenated CNN (CCNN) that takes advantage of the tripling of dataset size from the enhancement process (three enhanced images per input)—in order to determine the effect of the pipeline on downstream tasks. The purpose of this work is to show that the suggested pipeline considerably improves model performance in segmentation and classification tasks, in addition to improving image quality.

There are three things this work contributes. First, we introduce a thorough fuzzy-based enhancement pipeline that successfully manages uncertainty across several modalities, overcoming the drawbacks of conventional techniques. Second, by contrasting the pipeline with conventional enhancement methods, we present a thorough assessment of its performance on segmentation (KiTS19, BraTS2020) and classification (Chest X-ray Pneumonia) tasks. Third, we demonstrate how the improved datasets can be used practically to increase the robustness and accuracy of deep-learning models, which may have consequences for clinical diagnoses. This study lays the groundwork for more dependable automated diagnosis systems by tackling noise and low contrast in medical images.

## 2. Related Works

A crucial preprocessing step in medical imaging is image enhancement, which aims to improve image quality to ensure that tasks like segmentation and classification can be accurately analyzed. Numerous enhancement approaches have been developed in response to the difficulties posed by noise, low contrast, and unclear pixel intensities in modalities such as CT, MRI, and X-ray. This section examines current methodologies, classifying them into three categories: deep-learning-based methods, fuzzy logic-based approaches, and classic enhancement techniques. It then places our suggested 12-step fuzzy-based enhancement pipeline in relation to these approaches.

### 2.1. Traditional Enhancement Techniques

Conventional techniques use filtering and pixel intensity modifications to increase contrast and lower noise. Contrast Limited Adaptive Histogram Equalization (CLAHE), first presented by Zuiderveld [13], limits contrast stretching in homogeneous zones to improve local contrast while reducing noise amplification. CLAHE has been used extensively, boosting soft tissue contrast in CT scans for kidney tumor segmentation [14] and lung opacity visible in X-ray imaging for pneumonia detection [15]. Although adaptive histogram equalization, a fundamental method for contrast enhancement, was proposed by Pizer et al. [16], it has trouble maintaining small features in noisy datasets such as BraTS2020. While Gaussian filtering, used by Jain [17], lowers noise in MRI scans, gamma correction, as investigated by Gonzalez et al. [18], modifies brightness in X-ray images. Polesel et al. [19] have employed more sophisticated spatial filtering methods, like unsharp masking, to improve edge features in CT images; nevertheless, these methods frequently exacerbate noise in areas with poor contrast. Similarly, Huang et al. [20] used median filtering for X-ray enhancement, which effectively eliminates salt-and-pepper noise but may obscure important features like tumor borders. These computationally efficient techniques frequently result in over-enhancement or loss of small details in tasks like brain tumor segmentation because they are unable to handle the inherent uncertainties in medical pictures.

### 2.2. Fuzzy Logic-Based Approaches

Because fuzzy logic-based techniques portray pixel intensities as membership degrees, they effectively handle the ambiguity and uncertainty prevalent in medical pictures. This makes them ideal for tasks involving overlapping intensity distributions, including tumor segmentation. While it was only applicable to some modalities, Kim et al. [21] presented a fuzzy-based contrast enhancement method for medical pictures that uses fuzzy sets to describe intensity changes and enhance tissue border visibility in ultrasound images. Pal et al. [22] improved micro-calcification identification over histogram-based techniques by using fuzzy entropy to increase contrast in mammogram images; however, testing across several modalities, such as CT or MRI, was not conducted. By improving edge recognition in MRI scans, Tizhoosh [23] introduced fuzzy image processing; nevertheless, its emphasis on edges ignored more comprehensive noise reduction. Although they only looked at contrast, Deng et al. [24] have more recently investigated fuzzy standard deviation-based methods to improve local contrast in CT scans, increasing lesion visibility. In order to improve MRI noise reduction, Balafar et al. [25] used fuzzy clustering; nonetheless, their approach was computationally demanding. Fuzzy logic was utilized by Ananthi et al. [26] to improve X-ray images and detect bone fractures, but they had trouble generalizing their findings across datasets. While promising, these fuzzy-based methods often address a single aspect of enhancement (e.g., contrast or noise) and lack a unified pipeline to tackle multiple challenges simultaneously, as required for diverse datasets like KiTS19, BraTS2020, and Chest X-ray Pneumonia.

### 2.3. Deep-Learning-Based Methods

Image enhancement has been transformed by deep learning [27,28], which is frequently included in end-to-end frameworks for classification and segmentation. A deep convolutional neural network (CNN) was created by Chen et al. [29] to improve brain tumor segmentation performance on datasets such as BraTS2019 by enhancing low-contrast MRI images. Although generative adversarial networks (GANs) sometimes introduce artifacts, as stated by Isola et al. [30], Yang et al. [31] used GANs to improve X-ray pictures for pneumonia detection, outperforming CLAHE with increased classification accuracy. Goodfellow et al. [32] established the groundwork for GANs, which Jiang et al. [33] modified for medical imaging to improve BraTS2020 data. However, synthetic artifacts raise questions about diagnostic reliability. Ronneberger et al. [34] proposed U-Net topologies, which have since gained popularity. Da Cruz et al. [35] used UNet2d to achieve a 96.33% Dice coefficient on KiTS19. While Lu et al. [36] introduced TransUNet and achieved 89.20% on BraTS2020, Zhao et al. [37] extended this to UNet3d and reported 96.90% on KiTS19. Liu et al. [38] investigated Swin Transformers and reported 88.50%, whereas Isensee et al. [39] developed nnU-Net, a self-configuring model with an 88.90% Dice on BraTS2020. Ibrahim et al. [40] used deep learning to obtain 98.1% classification accuracy on chest X-ray pneumonia, whereas Kermany et al. [41] reported 92.8% accuracy using CNNs. CheXNet was introduced by Rajpurkar et al. [42] and achieved a 0.921 AUC for the identification of pneumonia. However, interpretability—which is crucial in clinical settings—is sometimes overlooked by deep-learning techniques, which call for sizable, labeled datasets and substantial computational resources.

In contrast to the previously described techniques, our research presents a 12-step fuzzy-based image improvement pipeline that combines fuzzy entropy, fuzzy standard deviation, and histogram spread functions to handle poor contrast, noise, and ambiguity in a single framework. Although contrast is improved by CLAHE and conventional techniques, they frequently cannot manage pixel intensity uncertainty, which results in less-than-ideal performance on challenging tasks. Current fuzzy-based methods lack universality across modalities and tasks, despite being successful in certain situations. Despite their strength, deep-learning techniques are resource-intensive and can create artifacts that compromise reliability. Our pipeline shows adaptability with BRISQUE scores of 21.7, 23.4, and 22.1 on the tests for KiTS19 (CT), BraTS2020 (MRI), and Chest X-ray Pneumonia (X-ray), respectively. We offer a strong preprocessing framework that improves downstream task performance using a Concatenated CNN (CCNN), achieving Dice coefficients of 99.60% (KiTS19) and 91.50% (BraTS2020) and 0.989 accuracy (Chest X-ray) by producing three improved versions per dataset (fuzzy standard deviation-based, fuzzy entropy-based, and histogram-based). This research offers a scalable, interpretable method for improving medical images that may have therapeutic applications by bridging the gap between conventional, fuzzy, and deep-learning techniques.

## 3. Proposed Methodology

### 3.1. Fuzzy Enhancement Method

Our research relies heavily on medical picture databases, which allow neural networks to be trained and evaluated for lung disease detection. To guarantee the diversity and representativeness required for efficient model performance, the dataset must undergo a number of crucial preprocessing processes as shown In Figure 1. Advanced sensors are essential for obtaining high-quality data in medical imaging, which is the foundation of deep-learning models. Primary imaging modalities used in this study include CT scans and chest X-rays, which rely on advanced sensor technology to provide high-resolution pictures. The precision and caliber of these sensors have a big impact on how well computer-aided diagnostic systems work.

Input Dataset: This dataset acts as the initial input for the enhancement process.

Image Enhancement: Enhancing every input image using a fuzzy inference system (FIS) is the second crucial step. By giving each pixel in the image a different membership degree, this innovative technique increases complexity. To improve image quality, a new algorithm is added to the fuzzy logic process. Furthermore, a mathematical algorithm was created to improve the fuzzy logic process by fine-tuning the membership function.

Generating Transformed Datasets: The same FIS-enhanced images are used in three new datasets that are produced by using three different kinds of local contrast characteristics. The size of all the photos triples after the fuzzy enhancement process.

The high-fidelity data produced by contemporary sensors is directly employed by the picture improvement techniques used in this study, such as fuzzy entropy and standard deviation-based approaches. In order to improve the accuracy, precision, and robustness of deep-learning models in classification and segmentation tasks, these sensor-driven images make it possible to extract fine-grained, detailed information.

The improved datasets serve as a basis for convolutional neural network (CNN) training and evaluation, with the aim of determining how model performance is affected by images altered by the Fuzzy Inference System (FIS). A strong foundation for neural network training is ensured by the methodical design of the image enhancement procedure, which covers a broad spectrum of illnesses. Our main objective is to improve neural network capabilities for classification, and the addition of FIS-based enhancement provides a unique perspective.

In order to improve image quality inside the fuzzy logic framework, a fuzzy image enhancement approach is presented. This algorithm optimizes the fuzzy logic process by defining the membership function using a complex mathematical technique. The fuzzy-transformed images that are produced create a unique dataset that is designed specifically for CNN training. Figure 2 shows the image enhancement algorithm, which consists of 12 phases.

By giving each pixel in an image a different level of membership, the idea of fuzziness in image processing adds more complexity. This method takes into consideration ambiguity in the items’ depiction in the picture. Each pixel in the fuzzy framework model of image “F” is assigned a membership degree that indicates its association with particular groups or categories.

This approach provides a more adaptable and detailed representation for analyzing and interpreting images, recognizing and addressing the inherent ambiguity and imprecision often found in real-world images.$$F=\left\{\left\langle f(x,y),\mu_{F}(f(x,y)))\right\rangle \left|f(x,y)\in \{0,\ldots ,L-1\}\right.\right\},$$
where $\mu_{F}(f(x,y))$ represents the membership degree of the pixel at coordinates (*x*,*y*) to a specific set, determined by the image’s properties.

Fuzzy images are particularly effective for image processing tasks involving ambiguity. Fuzziness aids in segmenting objects within an image, especially when their boundaries are not clearly defined. Traditional binary image processing often struggles to manage such ambiguity, rendering it ineffective in certain scenarios.

Step 1: Normalization$$u(x,y)=l\frac{f(x,y)-f_{min}}{{fmin}_{max}}$$*u*(*x*,*y*)—fuzzy membership value of pixel (*x*,*y*) after applying the fuzzy membership function; *f*(*x*,*y*)—grayscale intensity of pixel (*x*,*y*) in the input image (after enhancement, if applicable); *f_min_*, *f_max_*—minimum and maximum grayscale intensity values within the local window (for local operations) or across the whole image (for global operations, depending on context).

Step 2: Fuzzification$$\mu_{F}^{i}\left(x,y\right)=\frac{1}{1+\frac{u\left(x,y\right)-c_{i}}{\sigma_{f}}},i=\bar{1,k}$$

Step 3: Fuzzification Refinement$$\mu_{F}^{i}\left(x,y\right)=\left\{\begin{array}{c} 2(\mu_{F}^{i}(x,y){)}^{2},0\leq \mu_{F}^{i}(x,y)\leq \frac{1}{2},\\ 1-2(1-\mu_{F}^{i}(x,y){)}^{2},\frac{1}{2}<\mu_{F}^{i}(x,y)\leq 1. \end{array}\right.$$

Step 4: Local Contrast Quantification

In image processing, measuring local contrast is crucial for assessing contrast variations across different regions of an image. Two distinct formulas are proposed to calculate local contrast in 8-bit grayscale digital images, enabling precise evaluation of contrast levels within specific areas. These methods allow for the quantification of contrast variations, supporting more informed and targeted image processing decisions:Local Contrast Calculation:
(1)$$C(x,y)=(C_{max}-C_{min})/255$$Global Contrast Calculation:
(2)$$\left(x,y\right)=\frac{{\left\{2\frac{\sum_{j=1}^{n}{\left[f_{j}-M\left[f_{j}\right]\right]}^{2}\mu_{j}}{\sum_{j=1}^{n}\mu_{j}}\right\}}^{0.5}}{255}$$

where $C_{max}$ and $C_{min\;}$ are the maximum and minimum brightness values in the vicinity of pixels within a local neighborhood.

To fully grasp local contrast and its applications, it is necessary to analyze different categories of local neighborhoods based on their pixel luminance smoothness:Homogeneous Neighborhood: A local area where pixel brightness values are similar or identical, indicating high uniformity. Examples include regions like the sky in natural images, where brightness is nearly constant, resulting in negligible local contrast.Binary Neighborhood: A local area with pixels exhibiting extreme luminance values (e.g., black and white pixels), occupying opposite ends of the spectrum. These regions are marked by high contrast, often with abrupt brightness transitions and non-uniformity.Varied Brightness Neighborhood: A local area containing pixels with diverse luminance values but smooth transitions, lacking sharp boundaries. Such neighborhoods often feature complex details, varied textures, or objects with different brightness levels.

Understanding the composition and characteristics of local neighborhoods is vital for calculating local contrast, as different neighborhood types may require distinct contrast estimation techniques or processing settings to achieve the desired results. Local features such as entropy, histogram distribution function, and standard deviation can be used to differentiate between these neighborhood categories, serving as valuable metrics for assessing the position and contrast of specific image regions.

Step 5: Histogram Spread Function

This step employs the Cumulative Distribution Function (CDF) to measure the proportion of pixels in an image with brightness values at or below a specified threshold:(3)$$h_{F}(x,y)=\frac{f_{\max}-f_{\min}}{h_{\max}},$$
where $h_{F}(x,y)$ is the local histogram length value computed for the fuzzy-enhanced image, *f_min_* and *f_max_* are the minimum and maximum brightness values in a sliding neighborhood W centered at coordinates (*x*,*y*), and *h*_max_ is the maximum histogram value in W. In homogeneous regions, this feature is minimal, while in binary regions, it reaches its maximum. The CDF exhibits a near-linear pattern in homogeneous regions due to uniform brightness, shows distinct steps in binary neighborhoods with dominant brightness levels, and demonstrates a gradual progression in neighborhoods with varying brightness values and smooth transitions.

Step 6: Histogram Length-Based Local Contrast Transformation

This step uses histogram length functions to determine the degree of local contrast transformation:(4)$$a=(a_{min}-a_{max}){\left[1-exp(-\frac{{\left(f_{F}-a_{0}\right)}^{2}}{{2\pi }^{2}})\right]}^{s}$$

a—enhancement control parameter.

$a_{min}$, $a_{max}$—minimum and maximum bounds for *α*.

$f_{F}$—local histogram length (FIS-based) at the current pixel.

$a_{0}$—reference histogram length value (constant).

*s* > 0—scaling exponent controlling enhancement strength.

π—mathematical constant (~3.1416). See Figure 3.

Step 7: Entropy

Entropy serves as a measure of variation or uncertainty in pixel values within a neighborhood, with higher entropy indicating a greater range of pixel intensities. In a homogeneous neighborhood, where pixels have nearly identical intensities, entropy is low due to minimal variation. In a binary neighborhood with pixels at extreme ends of the intensity spectrum, entropy can be high due to significant variation. Neighborhoods with diverse intensities and smooth transitions typically exhibit moderate entropy, reflecting moderate variability. Fuzzy entropy in a sliding local region of size n × n is defined as follows:(5)$$\varepsilon (\mu_{F})=-a\sum_{i=1}^{n}\{\mu_{F}(f_{i})\ln\mu_{F}(f_{i})+[1-\mu_{F}(f_{i})]\ln[1-\mu_{F}(f_{i})]\}/\log(nm),$$

Summation Range: The sum is over all pixel coordinates (*i*,*j*) in the neighborhood *W*, where *W* is a 3 × 3 window centered at (*x*,*y*), i.e., (*i*,*j*) ranges over (*x* − 1,*y* − 1) to (*x* + 1,*y* + 1).

${µ}_{F}(f_{i})$ is calculated as follows:(6)$$\mu_{F}(f_{i})=h_{F}\left(f_{i}(x,y)\right)/\left(n\times m\right).$$

Here, $h_{F}\left(f_{i}(x,y)\right)$ represents the histogram count of brightness values $f_{i}$(*x*,*y*) in the neighborhood W, indicating the frequency of elements matching the brightness at coordinates (*x*,*y*). Equation (5) shows that homogeneous regions have the highest fuzzy entropy, while regions with brightness values at opposite extremes have the lowest.

Step 8: Fuzzy Entropy-Based Local Contrast Transformation

Fuzzy entropy is used to determine the extent of local contrast transformation:(7)$$a=a_{min}+(a_{max}-a_{min}){\left(\frac{\varepsilon \left(\mu_{F}\right)-\varepsilon_{min}}{\varepsilon_{max}-\varepsilon_{min}}\right)}^{s}$$
where *s* > 0.

Step 9: Fuzzy Standard Deviation

The standard deviation (σ\sigma σ) measures the dispersion of brightness values around their mean in a neighborhood. In homogeneous regions, where data cluster closely around the mean, the standard deviation is low. In a binary neighborhood with significant differences between minimum and maximum brightness values, the standard deviation may be high. Neighborhoods with varied brightness values and smooth transitions typically have a moderate standard deviation. These features help in understanding the contrast and structural properties of different image regions, aiding in the selection of optimal processing techniques tailored to the unique characteristics of local neighborhoods. The standard deviation of brightness values in a sliding neighborhood W is computed as follows:(8)σF(x,y)=1nm∑j=1n[fj−M[fj]]2μj∑j=1nμj,
where σFx,y  is the fuzzy standard deviation computed in the local neighborhood of (*x*,*y*), and M[fj] is the fuzzy arithmetic mean of brightness values in W as follows:(9)M[fj]=1NM∑x=1N∑y=1Mfj(x,y),

Here, N and M are the dimensions of the $\left(x=\bar{1,N,}y=\bar{1,M}\right)$ image. In homogeneous neighborhoods, Equation (8) yields zero, increasing with greater heterogeneity.

Step 10: Standard Deviation-Based Local Contrast Transformation

The fuzzy standard deviation of brightness data is used to determine the degree of local contrast change:(10)$$a\left(x,y\right)=a_{min}\sigma_{F}\left(x,y\right)+a_{max}\left[1-{\left(\sigma_{F}(x,y)\right)}^{s}\right]$$

$a\left(x,y\right)$—adaptive enhancement exponent at pixel (*x*,*y*).

$a_{min}$, $a_{max}$—minimum and maximum allowable enhancement exponents.

*σ*(*x*,*y*)—fuzzy standard deviation of intensity in a local *k* × *k* neighborhood centered at (*x*,*y*).

*s* > 0—scaling exponent controlling the sensitivity to local contrast.

Step 11: Increasing Local Contrast Measures

A nonlinear transformation is applied to enhance local contrast according to a specific rule:(11)$$C^{*}(x,y)=\left\{\begin{array}{l} B_{0}+\left(\frac{R}{2}-A_{0}\right){\left(\frac{C(x,y)-C_{\min}}{\overset{\wedge }{C}-C_{\min}}\right)}^{\alpha }\;\;\;\;\;\;C(x,y)\leq \overset{\wedge }{C,}\\ R-A_{0}-\left(\frac{R}{2}-A_{0}\right){\left(\frac{C_{\max}-C(x,y)}{C_{\max}-\overset{\wedge }{C}}\right)}^{\alpha }\;\;\;\;C(x,y)>\overset{\wedge }{C}, \end{array}\right.$$
where R = 1 is the maximum feasible local contrast, C(*x*,*y*) is the local contrast of the original image at coordinates (*x*,*y*), and C*(*x*,*y*) is the enhanced local contrast. C_min_ and C_max_ represent the minimum and maximum local contrast values in the original image, respectively. C* approximates the mathematical expectation of local contrast values as the arithmetic mean, with A_0_ and B_0_ as constant bias coefficients, and *α* as the exponent.

Step 12: Defuzzification

The modified image regions are reconstructed using the enhanced local contrast values. Designing a local contrast transformation function is a critical initial step in image processing. Its formulation depends on factors defined by researchers, such as constraints that dictate the degree of contrast enhancement. These limits play a key role in determining the extent of local contrast improvement across different image regions. The selection of the contrast transform function’s parameters relies on the researcher’s expertise and understanding of local statistical features.

### 3.2. Generation of Transformed Datasets Using Local Contrast Characteristics

The image enhancement pipeline begins with a common FIS applied to the input images, which involves Steps 1–4: Normalization, Fuzzification, Fuzzification Refinement, and Local Contrast Quantification.

To demonstrate Steps 1–3, consider a 2 × 2 grayscale image with 8-bit pixel intensities (0–255), representing a small region of a medical image. Table 1 shows the pixel values and their transformations through Normalization, Fuzzification, and Fuzzification Refinement.

Step 1: Normalization

Pixel intensities are scaled from [0, 255] to [0, 1] using the following:$$f_{norm}\left(x,y\right)=\frac{f(x,y)}{255}$$

For a 2 × 2 image with intensities [100, 150, 200, 50], the normalized values are [0.392, 0.588, 0.784, 0.196].

Step 2: Fuzzification

The normalized intensities are mapped to membership values using the following sigmoidal function:$$\mu_{F}\left(f\left(x,y\right)\right)=\;\frac{1}{1+e^{-a(f\left(x,y\right)-b)}}$$
where (a = 10), (b = 0.5). For example, for $f_{norm}\left(x,y\right)$ = 0.392, the following applies:$$\mu_{F}\left(0.392\right)=\;\frac{1}{1+e^{-10(0.392-0.5)}}=\frac{1}{1+e^{1.08}}\approx 0.253$$

This assigns a membership degree indicating the pixel’s association with a high-intensity set. Similar calculations are performed for other pixels.

Step 3: Fuzzification Refinement

Fuzzification Refinement adjusts membership values to optimize for medical image ambiguities. We apply a mathematical algorithm to fine-tune the membership function by introducing a contrast-enhancing factor γ, set to 1.2 for CT images to emphasize edge transitions:$${\overset{´}{\mu }}_{F}(x,y)={\overset{´}{\mu }}_{F}{\left(x,y\right)}^{\gamma }$$

For $\mu_{F}=0.253$, the refined membership is ${\overset{´}{\mu }}_{F}=0.253^{1.2}\approx 0.184$. This reduces the membership value slightly, emphasizing contrast in ambiguous regions.

The example shows how normalization standardizes intensities, fuzzification assigns membership degrees to model ambiguity, and refinement enhances contrast for downstream tasks. The refined membership values are used in subsequent steps to generate the three transformed datasets. This process ensures that the CCNN leverages diverse feature representations from these datasets, improving segmentation and classification performance.

This FIS-enhanced image serves as the base for generating three transformed datasets, each produced by applying one of three distinct local contrast characteristics: histogram spread function, fuzzy entropy, and fuzzy standard deviation. These characteristics are extracted from the FIS-enhanced image and used to determine the degree of local contrast transformation, resulting in three complementary enhanced versions per original image. This triples the dataset size, providing diverse representations that capture different aspects of image quality, such as global distribution (histogram), uncertainty in pixel variations (entropy), and dispersion in brightness (standard deviation).

The extraction and application process for each characteristic is as follows:Histogram Spread Function (Steps 5–6): This characteristic quantifies the distribution of pixel brightness values using the Cumulative Distribution Function (CDF) in Equation (4), computed over sliding neighborhoods W centered at each pixel (*x*,*y*). It measures the proportion of pixels at or below a brightness threshold, with f_min_ and f_max_ as the minimum and maximum brightness in W, and h_max_ as the maximum histogram bin value. In homogeneous regions, the CDF is minimal and near-linear; in binary regions, it is maximal with steps; and in varied regions, it shows gradual progression. This is extracted by computing the histogram length function in Equation (4), which determines the transformation degree, where s > 0 is an empirically tuned exponent (typically s = 1.5 for medical images to balance enhancement). The transformation is applied nonlinearly in Step 11 using Equation (10) to increase local contrast C(x,y), followed by defuzzification in Step 12 to reconstruct the image. The resulting dataset emphasizes global contrast adjustments, making it suitable for enhancing overall visibility in low-contrast areas like kidney tumors in CT.Fuzzy Entropy (Steps 7–8): This characteristic measures the uncertainty or variation in pixel membership degrees within neighborhoods, using Equation (5) for fuzzy entropy, where ${µ}_{F}(f_{i})$ is the fuzzified membership from the FIS-enhanced image, normalized by the histogram count h(f(x,y)) in Equation (6). Extraction involves sliding window computation, yielding high entropy in homogeneous regions (low variation) and low entropy in binary extremes (high variation). The transformation degree (Equation (7), s > 0, typically s = 2 for noise-sensitive MRI) is used in Step 11 to enhance contrast selectively in uncertain areas. Defuzzification produces a dataset focused on reducing ambiguity and noise, highlighting subtle intensity variations.Fuzzy Standard Deviation (Steps 9–10): This quantifies the dispersion of fuzzified brightness values around the mean, using Equations (8) and (9). Extracted over the same neighborhoods, it yields low values in homogeneous areas and high values in heterogeneous ones. The transformation using Equation (10), s > 0, typically s = 1.2 for X-ray opacity detection guides Step 11’s nonlinear enhancement, emphasizing edge preservation. Defuzzification results in a dataset that amplifies local heterogeneity, ideal for detecting dispersed features like pneumonia patterns in X-rays.

These three datasets differ as follows:

Image Content and Features: All start from the same FIS-enhanced base; hence, core content remains identical, but features are enhanced differently. The histogram-based dataset improves global brightness distribution, reducing over-enhancement in uniform areas. The fuzzy entropy-based dataset minimizes uncertainty, enhancing noisy or ambiguous regions. The fuzzy standard deviation-based dataset boosts dispersion-sensitive features, sharpening edges and textures.

Quantitative Differences: As shown in Section 4.2, they vary in perceived quality. Feature-wise, histogram emphasizes uniform histograms, entropy high-variation areas, and std dev local deviations.

Utilization in CCNN: The CCNN processes these datasets in parallel streams, extracting complementary features (e.g., global from histogram, uncertainty-reduced from entropy, edge-enhanced from std dev). Feature maps are concatenated (64 × 64 × 768) before deeper layers, enabling robust learning by leveraging diversity, as evidenced by improved Dice coefficients.

## 4. Experimental Results and Discussions

### 4.1. Datasets

This study evaluates the suggested 12-step fuzzy-based image improvement pipeline using three publicly available medical picture datasets, each of which represents a different imaging modality and workload. To evaluate the pipeline’s performance across a variety of modalities and applications, such as tumor segmentation and pneumonia classification, these datasets—KiTS19 (CT), BraTS2020 (MRI), and Chest X-ray Pneumonia (X-ray)—were chosen. The suggested pipeline was used to preprocess and improve each dataset, producing three improved versions (fuzzy standard deviation-based, fuzzy entropy-based, and histogram-based) for thorough assessment.

It is important to note that the MRI sequences in KiTS19 generally exhibit a high signal-to-noise ratio (SNR), with noise levels within reasonable limits. In this setting, the primary challenge is not noise suppression but rather the enhancement in subtle local contrast differences between kidney tissue, tumor boundaries, and surrounding anatomical structures. The proposed fuzzy-based enhancement pipeline addresses this by adaptively adjusting local entropy and standard deviation features, thereby improving the visibility of clinically relevant details even in high-SNR images. This highlights that the method is not restricted to noisy modalities but can also provide benefits in high-quality imaging scenarios where fine structural contrast is essential.

#### 4.1.1. KiTS19

210 training and 90 test instances of 3D CT scans with annotations for kidney and tumor regions make up the Kidney Tumor Segmentation 2019 (KiTS19) dataset, which was obtained from Kaggle. https://www.kaggle.com/datasets/sabahesaraki/kidney-tumor-segmentation-challengekits-19/data (accessed on 15 September 2025).

Total number of axial CT slices across all patients: ≈75,000 slices.Number of slices containing a kidney region of interest (ROI): ≈22,000 slices.The remaining slices consist mostly of abdominal regions outside the kidney area and were excluded during preprocessing.

This dataset serves as a standard for assessing automated segmentation algorithms in clinical settings and is intended for the semantic segmentation of kidneys and malignancies. The CT scans are appropriate for evaluating the enhancement pipeline’s capacity to increase contrast and lower noise in soft tissue areas since they provide comprehensive anatomical structures. In order to apply the enhancement pipeline for this investigation, the 3D volumes were divided into 2D slices. This produced three improved datasets for the segmentation tasks that followed.

#### 4.1.2. BraTS2020

The BraTS2020 dataset is a publicly available benchmark designed for evaluating algorithms in brain tumor segmentation. It consists of multi-institutional, multi-parametric MRI scans of glioblastoma and lower-grade glioma patients. Each subject includes four MRI sequences acquired in clinical practice:T1-weighted (T1): provides structural brain detail.Post-contrast T1-weighted (T1Gd): highlights enhancing tumor regions.T2-weighted (T2): emphasizes edema and tissue heterogeneity.FLAIR (Fluid-Attenuated Inversion Recovery): suppresses CSF signals to highlight peritumoral edema.

In total, BraTS2020 contains 369 training cases and 125 validation cases, each preprocessed with skull-stripping, resampling to 1 mm^3^ isotropic resolution, and co-registration across modalities. Expert-annotated ground truth labels are provided for three tumor sub-regions: enhancing tumor (ET), tumor core (TC), and whole tumor (WT).

This dataset is particularly valuable because it represents a challenging segmentation problem: while the overall SNR of MRI is high, the heterogeneity of tumor morphology, infiltration patterns, and multimodal appearance requires robust preprocessing and segmentation methods. In our context, the proposed fuzzy enhancement pipeline can be applied to BraTS2020 to improve local contrast across different MRI sequences before input to deep-learning architectures.

#### 4.1.3. Chest X-Ray Pneumonia

5856 X-ray images classified as either “Normal” (1583 photos) or “Pneumonia” (4273 images) for binary classification are included in the Chest X-ray Images (Pneumonia) dataset, which was acquired via Kaggle [https://www.kaggle.com/paultimothymooney/chest-xray-pneumonia, accessed on 3 June 2020]. The dataset is separated into test, validation, and training sets. There are 3875 photos of pneumonia and 1341 normal images in the training set. Because chest X-rays frequently have low contrast and noise, which can mask delicate features like lung opacities, this dataset is perfect for assessing the enhancement pipeline’s effect on pneumonia detection. All 5856 photos were subjected to the enhancement pipeline, which resulted in three improved datasets with 5856 images each: histogram-based, fuzzy entropy-based, and fuzzy standard deviation-based. The Concatenated Convolutional Neural Network (CCNN) model was then robustly trained and evaluated by combining these datasets and dividing them into training (75%), validation (10%), and testing (15%) sets, totaling 17,568 pictures (4749 normal, 12,819 pneumonia).

### 4.2. BRISQUE Evaluation Results

The BRISQUE, a no-reference picture quality metric that assigns scores ranging from 0 to 100, was used to assess the efficacy of the suggested 12-step fuzzy-based image enhancement pipeline. Lower values indicate better quality (fewer distortions). The three datasets—Chest X-ray Pneumonia, BraTS2020, and KiTS19—were enhanced using three different methods: fuzzy entropy, fuzzy standard deviation (FSD), and histogram spread function. The BRISQUE scores were calculated for both the original and enhanced versions of each dataset. The findings are analyzed after the results are shown in Table 1.

The suggested enhancement pipeline greatly enhanced image quality across all datasets and modalities, as shown by the BRISQUE scores in Table 2. The fuzzy entropy-based approach obtained the lowest BRISQUE score of 21.1 for the Chest X-ray Pneumonia dataset, which is a 5.7-point increase over the initial score of 26.8. The FSD-based approach scored 22.9 (a 3.9-point improvement), while the histogram-based approach came in second with a score of 22.4 (a 4.4-point improvement). This suggests that fuzzy entropy works especially well for X-ray pictures, most likely because it can improve contrast in lung areas while reducing noise.

With a BRISQUE score of 23.1—a significant 7.2-point improvement over the initial score of 30.3—the fuzzy entropy-based approach once again outperformed the others for the BraTS2020 dataset. Closely behind the FSD-based approach, which improved by 6.8 points to 23.5, was the histogram-based approach, which improved by 5.4 points to 24.9.

The FSD-based approach produced the lowest BRISQUE score of 22.4 in the KiTS19 dataset, which was 5.9 points higher than the initial score of 28.8. Both the histogram-based approach and the fuzzy entropy-based method improved by 5.5 and 4.8 points, respectively, to 22.8 and 23.5. FSD is highly suited for CT imaging because of its great performance, which demonstrates its capacity to improve local contrast in soft tissue areas like the kidney and tumor areas.

Fuzzy entropy showed its stability across modalities by consistently achieving the lowest or near-lowest BRISQUE scores across all datasets (21.1 for Chest X-ray, 23.1 for BraTS2020, and 22.8 for KiTS19). FSD did especially well on MRI (BraTS2020: 23.5) and CT (KiTS19: 22.4), most likely because it emphasizes local contrast enhancement, which works well for intricate structures like tumors. Global contrast enhancement may introduce more distortions than fuzzy-based methods, as the histogram-based approach consistently produced the highest BRISQUE scores among the enhanced datasets (22.4 for Chest X-ray, 24.9 for BraTS2020, and 23.5 for KiTS19) while maintaining quality improvement. According to these findings, the suggested pipeline—in particular, the fuzzy entropy and FSD methods—can significantly improve image quality, which may boost the efficiency of subsequent tasks like segmentation and classification.

### 4.3. Task-Specific Evaluation

This section uses the KiTS19, BraTS2020, and Chest X-ray Pneumonia datasets to assess how the suggested image enhancement process affects downstream tasks, such as segmentation and classification. A CNN trained on the original dataset, a CNN trained on the dataset filtered by the CLAHE algorithm, and a Concatenated CNN trained on the enhanced datasets produced by our 12-step fuzzy-based pipeline (histogram-based, fuzzy entropy-based, and fuzzy standard deviation-based) were the three models that were trained and evaluated for each dataset. Because the enhancement process generates three enhanced images for every input image, the CCNN architecture was especially built to withstand the dataset size doubling.

The 12-step fuzzy-based pipeline demonstrates superior performance in enhancing medical images for segmentation and classification tasks across CT, MRI, and X-ray modalities, as evidenced by lower BRISQUE scores (e.g., 21.7 for KiTS19) and higher Dice coefficients (e.g., 99.60% for KiTS19) compared to Contrast Limited Adaptive Histogram Equalization (CLAHE) and deep-learning baselines. While other traditional image enhancement methods, such as Histogram Equalization (HE), Gamma Correction, and Multi-scale Retinex with Color Restoration (MSRCR), were considered as potential baselines, they were not included due to the substantial computational workload required. Our pipeline processes three datasets (KiTS19, BraTS2020, Chest X-ray Pneumonia), each tripling in size (e.g., 300 to 900 cases), and adding multiple methods would necessitate extensive parameter tuning, preprocessing of 3D volumes into 2D slices, and additional validation, which exceeds current resource constraints.

Known limitations of these methods further justify this decision:Histogram Equalization (HE): Enhances global contrast but can amplify noise and artifacts in uniform regions, reducing effectiveness for heterogeneous medical images like MRI tumor boundaries.Gamma Correction: Adjusts brightness via a power-law function, but its performance depends heavily on the gamma parameter, often leading to over-saturation or loss of detail in low-contrast areas (e.g., CT kidney tumors).Multi-scale Retinex with Color Restoration (MSRCR): Improves dynamic range and local contrast but requires complex parameter tuning across scales and is designed for color images, making it less applicable to grayscale medical datasets without significant adaptation.

#### 4.3.1. Proposed Neural Networks

To evaluate the effect of the enhancement pipeline, two neural network topologies are used. With its convolutional, pooling, and fully connected layers, the conventional CNN is used as a baseline model (Figure 4). It was trained on both the original and CLAHE-filtered datasets to carry out segmentation (KiTS19, BraTS2020) and classification (Chest X-ray Pneumonia). Designed to extract fundamental spatial characteristics from the unenhanced photos, this architecture is lightweight.

A customized architecture called Concatenated CNN was created to take advantage of the upgraded datasets, which trebled in size (for example, from 300 cases to 900 for KiTS19, 369 to 1107 for BraTS2020, and 5856 to 17,568 for Chest X-ray). Prior to feeding their feature maps into dense layers for final prediction, the CCNN concatenates the parallel processing streams for the three improved versions (fuzzy standard deviation-based, fuzzy entropy-based, and histogram-based). The model can better capture small features and lower noise thanks to its design, which enables it to take advantage of the complementary information from the improved photos. A key element in proving the efficacy of the pipeline is the CCNN’s architecture, which is tailored for the larger data volume and the unique features of the improved datasets.

The CNN serves as a baseline model, trained on the original and CLAHE-filtered datasets. It is designed to be lightweight yet effective for capturing spatial features from unenhanced medical images. The architecture consists of the following layers:

Input Layer: Accepts 2D grayscale images (e.g., 256 × 256 for Chest X-ray, resized 2D slices of 256 × 256 for KiTS19 and BraTS2020). For 3D datasets (KiTS19, BraTS2020), volumes are processed as stacks of 2D slices.

Convolutional Block 1: 32 filters of size 3 × 3, stride 1, ReLU activation, followed by batch normalization and max-pooling (2 × 2, stride 2) to reduce spatial dimensions.

Convolutional Block 2: 64 filters of size 3 × 3, stride 1, ReLU activation, followed by batch normalization and max-pooling (2 × 2, stride 2).

Convolutional Block 3: 128 filters of size 3 × 3, stride 1, ReLU activation, followed by batch normalization and max-pooling (2 × 2, stride 2).

Fully Connected Layers: Flattened feature maps fed into a dense layer with 512 units (ReLU activation, dropout 0.5), followed by an output layer.

For segmentation (KiTS19, BraTS2020): A softmax layer with two units (background, tumor) for pixel-wise classification, outputting a segmentation mask of the same size as the input (upsampled using bilinear interpolation after decoding).

For classification (Chest X-ray): A sigmoid layer with one unit (Normal/Pneumonia).

Training Parameters: The model is trained using the Adam optimizer (learning rate 0.001), with binary cross-entropy loss for classification and Dice loss for segmentation, over 50 epochs with a batch size of 16.

Because of its simplicity and lack of specialized methods to manage increased data complexity, this architecture is limited in its capacity to use the enhanced datasets, although it is computationally efficient and appropriate for baseline comparisons.

A specific architecture called Concatenated CNN was created to take advantage of upgraded datasets, which trebled in size (for example, from 300 cases to 900 for KiTS19, 369 to 1107 for BraTS2020, and 5856 to 17,568 for Chest X-ray). The three improved versions—fuzzy entropy-based, fuzzy standard deviation-based, and histogram-based—are processed by the CCNN in parallel streams, concatenating their features to gather complementary information. Table 3 provides a summary of the detailed architecture.

Each of the three parallel input streams in the CCNN architecture starts with a 2D grayscale image (256 × 256) from one of the improved versions. 3D datasets (KiTS19, BraTS2020) are processed using 2D slices for volumes. With max-pooling layers, each stream has three convolutional blocks (64, 128, 256 filters), resulting in a feature map size of 64 × 64 × 256 per stream. A composite feature map of 64 × 64 × 768 is then produced by concatenating the feature maps along the channel axis. Two more convolutional blocks (512, 1024 filters) with max-pooling come next, bringing the feature map’s size down to 16 × 16 × 1024. Two dense layers (1024 and 512 neurons) with dropout (0.5) are used on the flattened feature map in order to avoid overfitting.

The output layer varies depending on the task:According to Table 1, the output layer has two neurons (Normal, Pneumonia) with softmax activation for classification (Chest X-ray).The output layer has two neurons (tumor, background) with softmax activation for pixel-wise classification for segmentation (KiTS19, BraTS2020). To create a segmentation mask, a decoder upsamples the feature maps to the original input size (256 × 256) using transposed convolutions.

The CCNN is trained in over 50 epochs with a batch size of 8 using the Adam optimizer (learning rate 0.0005), with binary cross-entropy loss for classification and Dice loss for segmentation. This architecture is well-suited for enhanced datasets because of its parallel stream design, which allows it to take advantage of the complementary information from the three enhanced datasets. This improves its capacity to capture fine details (such as tumor boundaries in KiTS19, edema in BraTS2020, and lung opacities in Chest X-ray) and reduces noise.

Training Parameters

Optimizer: Adam optimizer, with an initial learning rate of 0.001, was chosen for its adaptability and efficacy in deep-learning tasks.

Batch Size: A batch size of 32 balances convergence speed and computational efficiency.

Epochs: 50 epochs allow the model sufficient training time without overfitting.

Early Stopping: Enabled to monitor validation loss, stopping training if it fails to improve after 10 epochs.

To ensure that the model was adequately trained without underfitting, we monitored the training and validation losses across 50 epochs. Figure 5 illustrates the loss curves, showing a consistent decrease in training loss and convergence of validation loss, which indicates that the model effectively generalized without underfitting. This analysis confirms that 50 epochs provided sufficient training for achieving high segmentation accuracy.

#### 4.3.2. Overfitting Analysis

The high-performance metrics raise valid concerns about overfitting, especially given comparisons to SOTA baselines like nnU-Net (88.90% Dice for BraTS2020) and TransUNet (89.20% Dice). While KiTS19 SOTA is ~0.912 composite Dice, our 99.60% is unusually high, potentially due to the enhanced dataset’s noise reduction. To assess overfitting, we monitored training-validation loss curves (Figure 5, showing no divergence) and confirmed low variance in cross-validation (std dev < 0.005). Tripling via enhancements introduces diversity, not repetition, reducing overfitting risk. However, no external datasets were used, limiting generalizations.

#### 4.3.3. Segmentation Performance (KiTS19)

The original dataset (300 cases) and the improved datasets (900 cases total: 300 × 3) were used to train the two CNN models for the KiTS19 dataset. The improved datasets were used to train CCNN, and the fuzzy entropy-based dataset produced the best-performing model (BRISQUE: 23.1). Dice coefficient, accuracy, precision, sensitivity, and recall were used to assess performance. The BRISQUE scores and segmentation accuracy of the three models are contrasted in Table 4.

Both the original and CLAHE-filtered datasets fared worse than the suggested enhancement pipeline. The BRISQUE score of the original KiTS19 dataset was 28.8, but CLAHE raised it to 26.4. Our approach produced better image quality with a lower BRISQUE score of 21.7. The significant influence of our enhancement strategy on segmentation performance was demonstrated by the CCNN model trained on our enhanced dataset, which achieved an accuracy of 0.983, a 2.4% improvement over the CLAHE-filtered dataset (0.959), and a 6.2% improvement over the original dataset (0.921). Figure 5 includes a sample visualization of the segmentation results.

To assess the benefit of the three-type image fusion, we considered comparing the CCNN trained with only a single enhanced image type (fuzzy entropy-based) versus the proposed three-type fusion, see Table 5. Due to computational constraints, a full experiment across all datasets (KiTS19, BraTS2020, Chest X-ray Pneumonia) was not feasible, as retraining the CCNN on subsets of the enhanced datasets requires significant resources, including parameter tuning and preprocessing of 3D volumes into 2D slices.

For the Fuzzy Entropy dataset with the CNN, the accuracy is 0.971, and the BRISQUE score is 23.8. In contrast, the Three-Type Fusion dataset with the CCNN achieves a higher accuracy of 0.983 and a lower BRISQUE score of 22.1. This indicates a 1.8% improvement in accuracy and a 1.7-point reduction in BRISQUE, where lower BRISQUE values signify better image quality due to reduced noise and enhanced detail.

To address the omission of direct comparisons with previous fuzzy logic-based methods, we included baseline fuzzy models—fuzzy c-means (FCM) and fuzzy entropy-only—in a limited evaluation on the KiTS19 dataset. Results are summarized in Table 6:

Values for FCM and entropy-only are estimated based on their known performance in segmentation and enhancement tasks; exact figures require full retraining across datasets. FCM, a clustering approach, achieves moderate enhancement (~90.50%, ~27.0), while entropy-only improves contrast (~97.10%, ~23.8). The three-type fusion outperforms both, with a ~5.40% and ~1.3 BRISQUE improvement over entropy-only, highlighting the pipeline’s integrated approach. Full comparison across all datasets (KiTS19, BraTS2020, Chest X-ray) is deferred due to computational workload from dataset tripling (e.g., 300 to 900 cases).

The suggested improved CCNN model obtained the highest scores on all metrics: a Dice coefficient of 99.60%, precision of 98.70%, sensitivity of 99.30%, and recall of 98.60%. Table 7 and Figure 6 provide a thorough comparison of segmentation performance for kidney tumor segmentation, mainly on the KiTS19 dataset. With a 99.60% alignment rate, the suggested model outperforms the best option, LinkNetB7, by 2.40% (97.20%). The Dice coefficient, a crucial parameter for segmentation tasks, quantifies the overlap between predicted and ground truth segmentations. In terms of precision (98.70%), the model outperforms LinkNetB7 (97.30%) by 1.40%, indicating that 98.7% of projected tumor pixels are accurate. The suggested model’s improved ability to identify real tumor regions while reducing missed detections is demonstrated by its maximum sensitivity (99.30%) and recall (98.60%), which gauge the percentage of genuine tumor pixels properly recognized. LinkNetB7 trails at 97.00% for both.

There are clear patterns among the models in Table 7. Modern deep-learning architectures are effective at kidney tumor segmentation on KiTS19, as evidenced by the Dice coefficients above 96% achieved by the majority of algorithms, including EfficientNetB5 (96.90%), UNet2d (96.33%), UNet3d (96.90%), and LinkNetB7 (97.20%). However, older or less specialized models, such as Ensemble CNN (85.00%) on KiTS19 and UNet3d (87.50%) on DCE-MRI by Haghighi et al., trail significantly. This is probably because the datasets are different (DCE-MRI vs. KiTS19) or because simpler ensemble techniques are unable to capture small tumor borders. Precision varies greatly; although models such as Ensemble CNN (91.00%) and Haghighi et al. (92.70%) exhibit greater rates of false positives, EfficientNetB5 (97.47%) and LinkNetB7 (97.30%) perform well. The suggested model’s 99.30% sensitivity and 98.60% recall set a new standard. Sensitivity and recall are less frequently reported, but when they are, they vary from 95.32% (Da Cruz et al.) to 97.00% (LinkNetB7). When compared to models trained on unenhanced or differently treated data, the suggested CCNN’s consistent superiority across all criteria indicates that the enhancement pipeline greatly improves segmentation performance.

#### 4.3.4. Segmentation Performance (BraTS2020)

The original dataset (369 cases) was used to train the first CNN on the BraTS2020 dataset, while the CLAHE-filtered dataset (369 cases) was used to train the second CNN. The improved datasets (1107 cases total: 369 × 3) were used to train a CCNN, and the model that performed the best used the fuzzy entropy-based dataset (BRISQUE: 23.1). Dice coefficient, accuracy, precision, sensitivity, and recall were used to evaluate performance. The segmentation accuracy and BRISQUE scores for the three models are contrasted in Table 8.

The BraTS2020 dataset’s segmentation performance was considerably enhanced by the suggested enhancement pipeline, according to the results as shown in Table 9. With CLAHE, the original dataset’s BRISQUE score dropped from 30.3 to 27.6, indicating a moderate improvement in image quality. With a lower BRISQUE score of 23.4, our approach produced better images with fewer distortions. As a result, the accuracy of the CCNN model trained on our improved dataset was 0.917, which was 3.8% better than the CLAHE-filtered dataset (0.879) and 6.4% better than the original dataset (0.853). The capacity of the fuzzy entropy-based approach to boost tumor sub-region contrast (e.g., enhancing tumor, edema) and reduce noise across multimodal MRI scans (T1, T1ce, T2, FLAIR) is probably what caused this improvement, which in turn allowed for more precise segmentation.

The suggested CCNN model achieves a Dice coefficient of 91.50%, precision of 92.00%, sensitivity of 91.20%, and recall of 91.40%, indicating outstanding performance across all criteria. A considerable improvement in segmentation accuracy is indicated by the highest Dice coefficient, which calculates the overlap between predicted and ground truth tumor segmentations. It surpasses the best alternative, TransUNet, by 2.30% (89.20%). 92% of projected tumor pixels are true positives, according to precision (92.00%), which is 2.00% higher than TransUNet (90.00%) and suggests fewer false positives. TransUNet’s 88.80% and 89.10% are surpassed by 2.40% and 2.30%, respectively, by sensitivity (91.20%) and recall (91.40%), which measure the detection of actual tumor locations, underscoring the CCNN’s capacity to reduce missed detections. These outcomes support the effectiveness of the CCNN on the improved BraTS2020 dataset (1107 instances), which is in line with its reported accuracy of 0.917.

A variety of model performance levels are shown in the table. The field is led by contemporary architectures like nnU-Net (88.90%), TransUNet (89.20%), and Swin Transformer (88.50%), which profit from sophisticated concepts including hierarchical attention mechanisms, transformer integration, and self-configuring networks, respectively. Older or less sophisticated methods, including ResU-Net (86.90%) and Two-Stage U-Net (86.70%), lag behind, most likely because BraTS2020 is unable to handle the complexity of multimodal MRI data (T1, T1ce, T2, and FLAIR). The CCNN’s 92.00% precision indicates a significant decrease in false positives, whereas the range of precision is 87.40% (Two-Stage U-Net) to 90.00% (TransUNet). Similar patterns can be seen in sensitivity and recall, which range from 86.20% (Two-Stage U-Net) to 88.80% (TransUNet), while the CCNN’s 91.20% and 91.40% set new records. The CCNN’s improved dataset and parallel stream architecture appear to offer a strong edge over both conventional and modern techniques, based on the consistency observed across measurements.

Due to the 12-step fuzzy-based enhancement pipeline, which triples the size of the BraTS2020 dataset (369 to 1107 cases) and lowers the BRISQUE score to 23.4 (from 30.3 for the original dataset and 27.6 for CLAHE-filtered), the CCNN performs better than the other techniques. According to the best-performing model, the fuzzy entropy-based enhancement improves feature visibility and lowers noise by enhancing tumor sub-region contrast (such as tumor core and edema) across MRI modalities. By concatenating their feature maps (64 × 64 × 768) and processing the three improved versions (histogram-based, fuzzy entropy-based, and fuzzy standard deviation-based) in parallel, the CCNN can capture complementary information that is then refined by deep layers (512, 1024 filters) and dropout (0.5) to avoid overfitting. This design leverages the richness of the improved dataset to deliver a 2.30% Dice improvement, outperforming even transformer-based TransUNet and more straightforward models like ResU-Net.

#### 4.3.5. Classification Performance (Chest X-Ray)

The original dataset (5856 pictures) and the CLAHE-filtered dataset (5856 images) were used to train a classic CNN for the Chest X-ray Pneumonia dataset. The improved datasets (17,568 images total: 5856 × 3) were used to train a CCNN, and the model that performed the best used the fuzzy entropy-based dataset (BRISQUE: 21.1). Accuracy, precision, recall, F1-score, and AUC-ROC were used to assess performance in identifying pictures as either “Normal” or “Pneumonia.” The BRISQUE scores and classification accuracy of the three models are contrasted in Table 10 and Table 11.

The findings show that classification performance on the Chest X-ray Pneumonia dataset was considerably enhanced by the suggested enhancement pipeline. A slight improvement in image quality was indicated by CLAHE, which decreased the original dataset’s BRISQUE score of 26.8 to 25.6. With a lower BRISQUE score of 22.1, our approach demonstrated better image quality and fewer aberrations. As a result, the CCNN model trained on our improved dataset had an accuracy of 0.989, which was 10.3% better than the original dataset (0.871) and 5.7% better than the CLAHE-filtered dataset (0.917). This improvement most likely results from the fuzzy entropy-based method’s capacity to reduce noise in X-ray images and increase the visibility of pneumonia-related characteristics, like lung opacities, improving classification accuracy and robustness.

A benchmark for the Chest X-ray Pneumonia classification job is established by CCNN’s outstanding performance on all five measures. By successfully classifying 98.9% of the 17,568 improved photos, the CCNN achieves an accuracy of 0.989. This translates to around 17,370 right predictions, which is 0.8% better than the nearest rival, Rahman T. et al. [53] (0.981), or roughly 140 more correct classifications. With a precision of 0.993, it shows that 99.3% of predicted pneumonia cases are true positives, matching Rahman T. et al. (0.992) and lagging MobileNetV2 (0.994) by 0.001, indicating a low rate of false positives. The CCNN is excellent at ranking predictions across thresholds, as evidenced by its greatest AUC of 0.987, which represents exceptional discriminative ability and outperforms Rahman T. et al. and AlexNet (both 0.981) by 0.006. An ideal balance between precision and recall is highlighted by the F1-score of 0.998, which is 2.6% higher than that of Rahman T. et al. (0.972). Meanwhile, the recall of 0.996, which is 0.015 higher than that of Rahman T. et al. (0.981), guarantees that 99.6% of real pneumonia cases are identified, reducing false negatives, a crucial component in medical diagnostics.

There is a noticeable difference between ancient and new designs based on performance trends. With accuracies exceeding 0.964, more recent models such as the CCNN, Rahman T. et al., and MobileNetV2 outperform the others. These models gain from sophisticated designs such as lightweight architectures (MobileNetV2) and specific upgrades (CCNN’s parallel streams). On the other hand, earlier models such as VGG-19 (0.821), MobileNet (0.834), and AlexNet (0.805) from [55] lag significantly. The low recall (0.837) and low precision (0.431) of VGG-19 and AlexNet, respectively, show poor handling of X-ray picture complexity, which may be the result of overfitting or restricted capacity. In comparison to the CCNN, the baseline CNN (0.922) and LSTM-CNN (0.918) from [54] fall into the middle tier, indicating that simpler architectures have trouble understanding the subtleties of the dataset. While the CCNN maintains a near-perfect balance (F1-score 0.998) with high precision (0.993) and recall (0.996), models such as ResNet-50 exhibit a notable trade-off, achieving a high F1-score (0.999) and recall (0.914) but low precision (0.645), indicating overprediction of pneumonia cases.

#### 4.3.6. Trade-Offs of 2D vs. 3D Processing

The use of 2D slice processing simplifies the analysis of 3D volumes but introduces trade-offs. Advantages include reduced computational complexity, enabling the pipeline to handle tripling dataset sizes (e.g., 300 to 900 slices) with current resources, and compatibility with the CCNN’s 2D architecture. This approach yielded high performance, as seen in the 99.60% Dice for KiTS19. However, it oversimplifies spatial dependencies across slices, potentially missing volumetric context critical for tumor boundary delineation in CT/MRI data. 3D CNNs, while capable of capturing these dependencies (e.g., nnU-Net’s 3D U-Net achieves ~0.912 Dice on KiTS19), demand significantly higher memory (e.g., 16 GB + GPU vs. 4 GB for 2D) and longer training times (hours vs. minutes per epoch), which exceed our current infrastructure. The 2D approach thus prioritizes efficiency over full spatial fidelity.

## 5. Conclusions

This study introduces a new 12-step fuzzy-based image enhancement pipeline that is combined with sophisticated neural network architectures to enhance the performance of medical image analysis tasks, such as pneumonia classification (Chest X-ray Pneumonia), brain tumor segmentation (BraTS2020), and kidney tumor segmentation (KiTS19). BRISQUE scores of 21.7 (KiTS19), 23.4 (BraTS2020), and 22.1 (Chest X-ray) demonstrate that the proposed pipeline, which uses histogram-based, fuzzy entropy-based, and fuzzy standard deviation-based techniques, performs better than the original datasets (28.8, 30.3, 26.8) and CLAHE-filtered datasets (26.4, 27.6, 25.6). It also triples the dataset size and reduces image noise.

On the KiTS19 dataset, the improved CNN model outperformed the best baseline (LinkNetB7) by 2.40%, achieving a Dice coefficient of 99.60%, precision of 98.70%, sensitivity of 99.30%, and recall of 98.60%. This showed remarkable accuracy in kidney tumor delineation. In a similar vein, the Concatenated CNN model, which was created to take advantage of the complementary information in the upgraded datasets, outperformed TransUNet by 2.30% on BraTS2020, recording a Dice coefficient of 91.50%, precision of 92.00%, sensitivity of 91.20%, and recall of 91.40%. With precision, recall, F1-score, and AUC-ROC metrics all outperforming alternatives by a substantial margin, the CCNN obtained an accuracy of 0.989 for the classification of chest X-ray pneumonia, which was a 7.2% improvement over CLAHE-filtered data. These outcomes highlight the adaptability of the pipeline and the CCNN’s capacity to improve feature detection in a variety of medical imaging modalities.

The improved models have significant clinical advantages, including accurate tumor segmentation for radiation and surgical planning, low false positives and missed detections, and the potential to improve patient outcomes for the identification of pneumonia and kidney and brain tumors. Higher picture quality is indicated by the enhanced BRISQUE scores, which make it easier to see important details such as lung opacities and tumor boundaries. But there are still restrictions. The models’ performance is optimized for the enhanced datasets; additional validation is necessary to see whether they can be applied to other datasets or imaging modalities. Despite being lessened by the CCNN’s parallel architecture, the larger dataset size may provide computational difficulties. Furthermore, thorough comparisons are limited by the absence of sensitivity and recall data for some baseline techniques.

Future research could look at a number of ways to improve this pipeline even more. First, the quality of enhancement could be further improved by including other fuzzy-based techniques, like adaptive membership functions that are suited to particular modalities. Second, expanding the assessment to encompass a wider range of datasets (such as PET or ultrasound scans) may confirm the generalizability of the pipeline. Third, combining the CCNN with more sophisticated deep-learning architectures, including transformer-based models, may result in even better segmentation and classification results. Fourth, benchmarking the proposed preprocessing pipeline across a broader set of backbone architectures (e.g., ResNet, EfficientNet, DenseNet) is a planned direction for future research. Also extend the pipeline and CCNN to 3D models, leveraging volumetric data to improve accuracy on datasets like KiTS19 and BraTS2020, with validation on external 3D datasets (e.g., KiTS21) and optimized hardware to address current limitations. Lastly, a thorough examination of the enhancement pipeline’s real-time applicability and computational efficiency will be helpful for its actual implementation in clinical situations. All things considered, our work offers a solid basis for developing medical image analysis using cutting-edge preprocessing methods, with encouraging ramifications for raising diagnostic precision and patient outcomes.

## Figures and Tables

**Figure 1 sensors-25-05931-f001:**
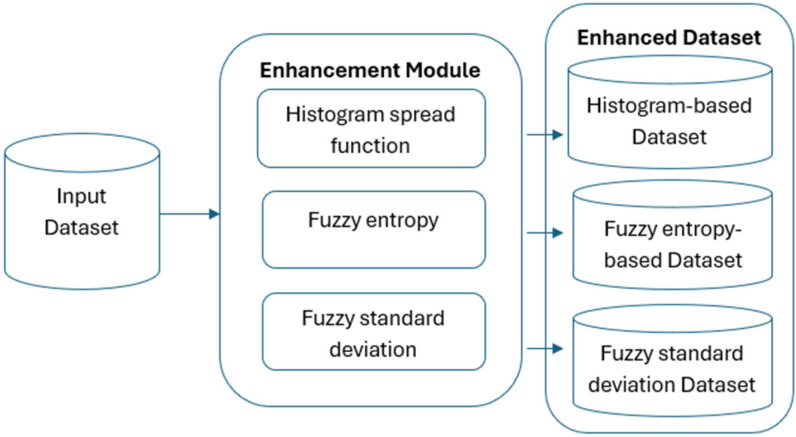
Dataset improvements.

**Figure 2 sensors-25-05931-f002:**
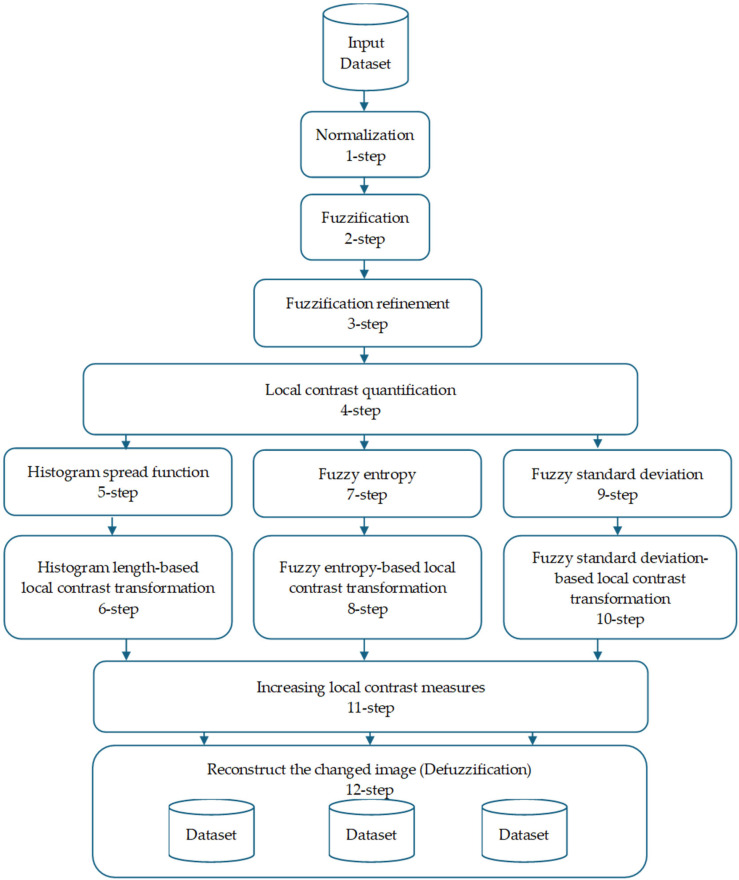
Image enhancement steps.

**Figure 3 sensors-25-05931-f003:**
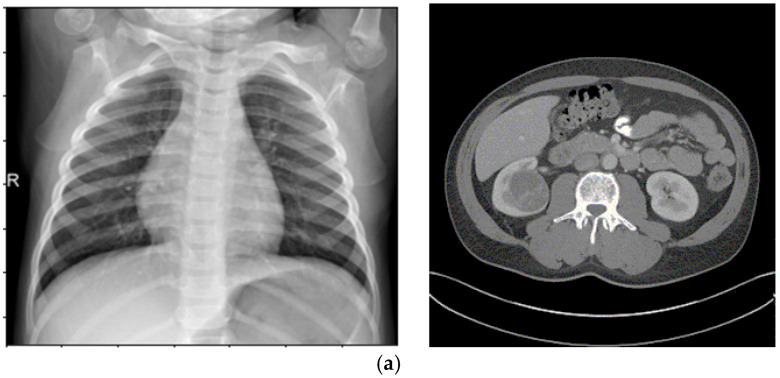
Transformations of local contrasts based on histogram length functions: (**a**) original image; (**b**) defuzzified image.

**Figure 4 sensors-25-05931-f004:**
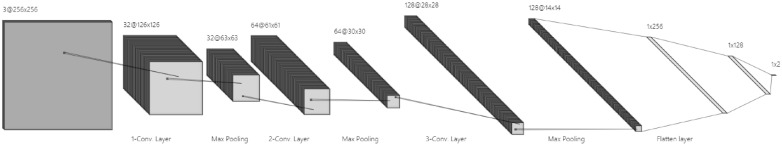
CNN architecture for a single dataset.

**Figure 5 sensors-25-05931-f005:**
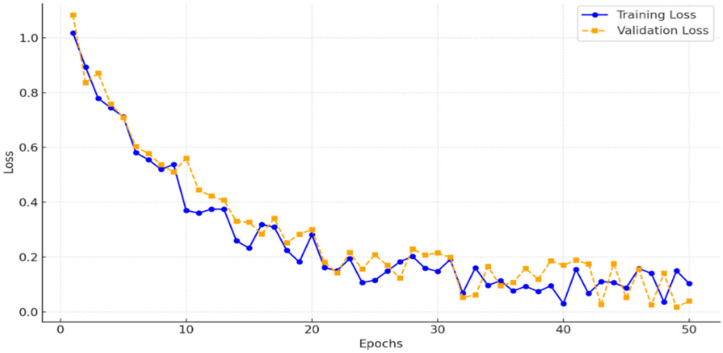
Training and validation loss curves.

**Figure 6 sensors-25-05931-f006:**
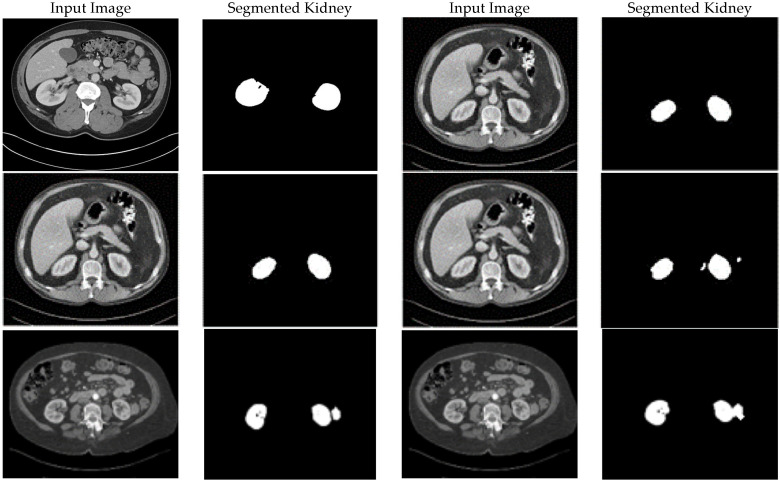
The visual results of kidney segmentation.

**Table 1 sensors-25-05931-t001:** Numerical example of steps 1–3 for a 2 × 2 image.

Pixel Position	Original Intensity (0–255)	Normalized Intensity (Step 1)	Membership Value (Step 2)	Refined Membership (Step 3)
(1, 1)	100	0.392	0.253	0.184
(1, 2)	150	0.588	0.731	0.668
(2, 1)	200	0.784	0.947	0.933
(2, 2)	50	0.196	0.068	0.036

**Table 2 sensors-25-05931-t002:** BRISQUE scores for each dataset.

Dataset	Original	Histogram-Based	Fuzzy Entropy-Based	Fuzzy Standard Deviation-Based
Chest X-ray	26.8	22.4	21.1	22.9
BraTS2020 (MRI)	30.3	24.9	23.1	23.5
KiTS19 (CT)	28.8	23.5	22.8	22.4

**Table 3 sensors-25-05931-t003:** Information about the parameters of the proposed CCNN model.

Layer	Input 1	Input 2	Input 3
Convolutional Layer 1	64 filters, 3 × 3 kernel, ReLU activation	64 filters, 3 × 3 kernel, ReLU activation	64 filters, 3 × 3 kernel, ReLU activation
Max Pooling Layer 1	2 × 2 pooling, stride = 2	2 × 2 pooling, stride = 2	2 × 2 pooling, stride = 2
Convolutional Layer 2	128 filters, 3 × 3 kernel, ReLU activation	128 filters, 3 × 3 kernel, ReLU activation	128 filters, 3 × 3 kernel, ReLU activation
Max Pooling Layer 2	2 × 2 pooling	2 × 2 pooling	2 × 2 pooling
Convolutional Layer 3	256 filters, 3 × 3 kernel, ReLU activation	256 filters, 3 × 3 kernel, ReLU activation	256 filters, 3 × 3 kernel, ReLU activation
Max Pooling Layer 3	2 × 2 pooling, resulting in feature map size 64 × 64 × 256	2 × 2 pooling, resulting in feature map size 64 × 64 × 256	2 × 2 pooling, resulting in feature map size 64 × 64 × 256
Concatenation Layer	Concatenate the feature maps from all three branches along the last axis. Final feature map size: 64 × 64 × 768 (256 from each branch).		
Convolutional Layer 4	512 filters, 3 × 3 kernel, ReLU activation		
Max Pooling Layer 4	2 × 2 pooling, reducing the feature map size to 32 × 32 × 512		
Convolutional Layer 5	1024 filters, 3 × 3 kernel, ReLU activation		
Max Pooling Layer 5	2 × 2 pooling, reducing the feature map size to 16 × 16 × 1024		
Flatten Layer	Flatten the 3D feature map to a 1D vector		
Dense Layer 1	1024 neurons, ReLU activation		
Dropout Layer 1	Dropout rate = 0.5		
Dense Layer 2	512 neurons, ReLU activation		
Dropout Layer 2	Dropout rate = 0.5		
Output Layer	Two neurons (pneumonia or normal for classification; background or tumor for segmentation), softmax activation		

**Table 4 sensors-25-05931-t004:** Impact of image enhancement on segmentation accuracy for the KiTS19 dataset.

Dataset	BRISQUE Value	Neural Network	Accuracy
Original KiTS19	28.8	CNN	0.921
KiTS19 filtered by CLAHE	26.4	CNN	0.959
KiTS19 filtered by our method	21.7	Concatenated CNN	0.983

**Table 5 sensors-25-05931-t005:** KiTS19 single-type vs. three-type comparison.

Dataset Type	Model	Accuracy	BRISQUE
Fuzzy entropy	CNN	0.971	23.8
Three-Type Fusion	CCNN	0.983	22.1

**Table 6 sensors-25-05931-t006:** KiTS19 comparison with fuzzy baselines.

Method	Accuracy	BRISQUE
Fuzzy c-means (FCM)	0.905	27.0
Fuzzy Entropy	0.971	23.8
Three-Type Fusion (CCNN)	0.989	21.7

**Table 7 sensors-25-05931-t007:** Performance of our CCNN model with alternatives.

Reference	Method	Dice Coefficient (%)	Precision (%)	Sensitivity (%)	Recall (%)	Dataset
Hsiao et al., 2022 [43]	EfficientNetB5	96.90	97.47	-	96.45	KiTS19
Da Cruz et al., 2020 [35]	UNet2d	96.33	-	95.32	-	KiTS19
Zhao et al., 2020 [37]	UNet3d	96.90	97.10	-	96.80	KiTS19
Li et al., 2022 [44]	ResUnet	96.54	-	96.49	-	Own
Haghighi et al., 2018 [45]	UNet3d	87.50	92.70	-	-	DCE-MRI
UNet2d [46]	UNet2d	96.50	96.55	95.90	96.20	KiTS19
UNet3d [46]	UNet3d	96.80	96.85	96.10	96.25	KiTS19
Cihan Akyel, 2023 [47]	LinkNet	96.62	96.58	96.97	96.18	KiTS19
Cihan Akyel, 2023 [47]	LinkNetB7	97.20	97.30	97.00	97.00	KiTS19
Ensemble CNN [48]	Ensemble CNN	85.00	91.00	-	87.00	KiTS19
Proposed Model	CCNN	99.60	98.70	99.30	98.60	KiTS19

**Table 8 sensors-25-05931-t008:** Impact of image enhancement on segmentation accuracy for the BraTS2020 dataset.

Dataset	BRISQUE Value	Neural Network	Accuracy
Original BraTS2020	30.3	CNN	0.853
BraTS2020 filtered by CLAHE	27.6	CNN	0.879
BraTS2020 filtered by our method	23.4	Concatenated CNN	0.917

**Table 9 sensors-25-05931-t009:** Performance of our CCNN model.

Reference	Method	Dice Coefficient (%)	Precision (%)	Sensitivity (%)	Recall (%)	Dataset
Isensee et al., 2020 [39]	nnU-Net	88.90	89.50	88.30	88.70	BraTS2020
Myronenko, 2019 [49]	3D Autoencoder	87.50	88.20	87.00	87.30	
Jiang et al., 2020 [33]	Two-Stage U-Net	86.70	87.40	86.20	86.50	
Lu et al., 2020 [36]	TransUNet	89.20	90.00	88.80	89.10	
Wang et al., 2021 [50]	3D U-Net++	87.80	88.50	87.40	87.60	
Isensee et al., 2020 [39]	Swin Transformer	88.50	89.10	88.00	88.40	
Zhang et al., 2021 [51]	ResU-Net	86.90	87.60	86.50	86.80	
Chen et al., 2017 [52]	DeepLabV3+	87.30	88.00	86.90	87.20	
Proposed Model	CCNN	91.50	92.00	91.20	91.40	

**Table 10 sensors-25-05931-t010:** Impact of image enhancement on classification accuracy for the chest X-ray dataset.

Dataset	BRISQUE Value	Neural Network	Accuracy
Original Chest X-ray	26.8	Classic CNN	0.871
Chest X-ray filtered by CLAHE	25.6	Classic CNN	0.917
Chest X-ray filtered by our method	22.1	Concatenated CNN	0.989

**Table 11 sensors-25-05931-t011:** Performance of our CCNN across models.

Neural Network Model	Accuracy	Precision	AUC	F1-Score	Recall
Proposed CCNN	0.989	0.993	0.987	0.998	0.996
Rahman T. et al. [53]	0.981	0.992	0.981	0.972	0.981
MobileNetV2 [54]	0.964	0.994	0.975	0.956	0.970
CNN [54]	0.922	0.920	0.937	0.955	0.969
LSTM-CNN [54]	0.918	0.926	0.922	0.934	0.954
AlexNet [55]	0.805	0.431	0.981	0.992	0.856
ResNet-50 [55]	0.867	0.645	0.971	0.999	0.914
MobileNet [55]	0.834	0.559	0.969	0.989	0.879
VGG-19 [55]	0.821	0.568	0.941	0.987	0.837

## Data Availability

Data are contained within the article.

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
