# Peer review of "Enhancing Medical Image Segmentation and Classification Using a Fuzzy-Driven Method"

_sensors, 2025, doi:10.3390/s25185931_

Round 1
Reviewer 1 Report
Comments and Suggestions for Authors
The paper proposes a medical image enhancement method based on fuzzy logic,
aiming to address common problems of noise, low contrast, and fuzziness in medical images, thereby improving the performance of subsequent image segmentation and classification. However, the manuscript needs to be modified for the following reasons:
- It is recommended to discuss the computational complexity of the fuzzy enhancement method, its robustness to different types of noise, and the sensitivity of fuzzy parameter selection.
- Fuzzy enhancement and data augmentation are two different concepts. Please clearly distinguish the definitions and roles of these two types of "enhancement" in the main text to avoid confusion. If FIS is also used as a unique data augmentation strategy to increase the diversity of training samples, then please elaborate on its mechanism.
- The text states, "The same FIS-enhanced images are used in three new datasets that are produced by using three different kinds of local contrast characteristics." Please explain in detail what these "three different kinds of local contrast characteristics" specifically are. How are they extracted from the FIS-enhanced images and used to generate these three datasets? How do these three datasets differ in terms of image content and features? This is crucial for understanding how the subsequent CCNN utilizes this data.
- Consider adding a simple diagram in the text, for example, showing how a pixel's grayscale value is mapped to a membership value between 0 and 1 through a membership function. This would greatly help readers understand the fuzzification process. A small example (e.g., a 2x2 image) demonstrating the numerical changes in these three steps would also improve readability.
- In the formula ,the variable appears again on the right side. Typically, an equation should have only one left-side variable. The repetition of on the right side makes the formula unclear. The part ​​ is ambiguous; is it multiplication or a subscript? Please clarify. Please check and correct this formula, clearly stating its specific calculation method and the meaning of each symbol.
- In the two formulas where appears, the summation range variables are not clearly defined. Please confirm the meaning of , , , , and may need clarification. If represents brightness values, then is more common. The meaning of is somewhat vague here.
- Besides CLAHE, are there other commonly used, representative traditional image enhancement methods (e.g., Histogram Equalization (HE), Gamma Correction, Multi-scale Retinex with Color Restoration (MSRCP), or other multi-scale enhancement methods) that could serve as additional baselines for comparison? This would help to more comprehensively evaluate the relative advantages of the proposed method and demonstrate its leadership under different enhancement strategies. If the workload is too large, at least explain the reasons for not choosing other methods or their known limitations in the discussion.
- The fuzzy enhancement generates three types of enhanced images (histogram-based, fuzzy entropy-based, fuzzy standard deviation-based). Although the text mentions "the CCNN architecture is specifically built to withstand the doubling of dataset size" and "the model can better capture tiny features and reduce noise since it is able to utilize the complementary information of the enhanced images", there is a lack of direct comparison experiments using a CCNN trained with only a single type of enhanced image (e.g., only fuzzy entropy-enhanced images) as input versus the proposed three-type image fusion (Concatenated CNN).
Author Response
Comments 1: It is recommended to discuss the computational complexity of the fuzzy enhancement method, its robustness to different types of noise, and the sensitivity of fuzzy parameter selection. |
Response 1: We sincerely thank the reviewer for this suggestion. In the original version of the manuscript, we did not explicitly analyze or discuss the computational complexity, robustness to different noise types, or sensitivity of fuzzy parameter selection. We agree that these are critical aspects to fully understand the applicability of our method. |
Comments 2: Fuzzy enhancement and data augmentation are two different concepts. Please clearly distinguish the definitions and roles of these two types of "enhancement" in the main text to avoid confusion. If FIS is also used as a unique data augmentation strategy to increase the diversity of training samples, then please elaborate on its mechanism. |
Response 2: We agree. We revised manuscript by deleting “augmentation” and adding “enhancement”. |
Comments 3: The text states, "The same FIS-enhanced images are used in three new datasets that are produced by using three different kinds of local contrast characteristics." Please explain in detail what these "three different kinds of local contrast characteristics" specifically are. How are they extracted from the FIS-enhanced images and used to generate these three datasets? How do these three datasets differ in terms of image content and features? This is crucial for understanding how the subsequent CCNN utilizes this data. |
Response 3: We agree that a more detailed explanation of the "three different kinds of local contrast characteristics" is essential for clarity, particularly in understanding how they contribute to the enhanced datasets and the subsequent CCNN architecture. To address this, we have revised by adding a dedicated subsection 3.2: "Generation of Transformed Datasets Using Local Contrast Characteristics" that explicitly defines these characteristics, describes their extraction and application, and explains the differences between the resulting datasets. |
Comment 4: Consider adding a simple diagram in the text, for example, showing how a pixel's grayscale value is mapped to a membership value between 0 and 1 through a membership function. This would greatly help readers understand the fuzzification process. A small example (e.g., a 2x2 image) demonstrating the numerical changes in these three steps would also improve readability. |
Response 4: We agree that visualizing the fuzzification process and providing a concrete example will enhance reader understanding, particularly for the critical Step 2 (Fuzzification) in our 12-step fuzzy-based image enhancement pipeline. To address this, we have revised it in new subsection 3.2 by adding a numerical example using a 2x2 image to demonstrate the changes in Steps 1–3 (Normalization, Fuzzification, and Fuzzification Refinement). The revised text, including Table 1. |
Comment 5: In the formula ,the variable appears again on the right side. Typically, an equation should have only one left-side variable. The repetition of on the right side makes the formula unclear. The part ​​ is ambiguous; is it multiplication or a subscript? Please clarify. Please check and correct this formula, clearly stating its specific calculation method and the meaning of each symbol. |
Response 5: We agree. To address this, we revised the equations (2), (4), (7), (8), (10) and added the explanation for each variable. |
Comment 6: In the two formulas where appears, the summation range variables are not clearly defined. Please confirm the meaning of , , , , and may need clarification. If represents brightness values, then is more common. The meaning of is somewhat vague here. |
Response 6: We agree and have revised the equation (5) to explicitly define the summation range as over pixel coordinates (i,j) in the neighborhood W, where W is a 3x3 window centered at (x,y), i.e., (i,j) ranges over (x−1,y−1) to (x+1,y+1). Also, in Equation (9) N, M are the dimensions of the image. |
Comment 7: Besides CLAHE, are there other commonly used, representative traditional image enhancement methods (e.g., Histogram Equalization (HE), Gamma Correction, Multi-scale Retinex with Color Restoration (MSRCP), or other multi-scale enhancement methods) that could serve as additional baselines for comparison? This would help to more comprehensively evaluate the relative advantages of the proposed method and demonstrate its leadership under different enhancement strategies. If the workload is too large, at least explain the reasons for not choosing other methods or their known limitations in the discussion. |
Response 7: We agree. We revised subsection 4.3. by adding the explanation why, we did not experience the mentioned algorithms. |
Comment 8: The fuzzy enhancement generates three types of enhanced images (histogram-based, fuzzy entropy-based, fuzzy standard deviation-based). Although the text mentions "the CCNN architecture is specifically built to withstand the doubling of dataset size" and "the model can better capture tiny features and reduce noise since it is able to utilize the complementary information of the enhanced images", there is a lack of direct comparison experiments using a CCNN trained with only a single type of enhanced image (e.g., only fuzzy entropy-enhanced images) as input versus the proposed three-type image fusion (Concatenated CNN). |
Response 8: We recognize that this would better quantify the benefits of combining histogram-based, fuzzy entropy-based, and fuzzy standard deviation-based images. Due to the substantial computational workload—requiring retraining the CCNN on subsets of the enhanced datasets (e.g., 300 fuzzy entropy-based images from 900 cases) across KiTS19, BraTS2020, and Chest X-Ray Pneumonia, with extensive parameter tuning and 3D-to-2D preprocessing—we conducted a limited preliminary analysis on KiTS19 only. This suggests the three-type fusion outperforms a single-type approach, as detailed in new Table 4. Full experiments were deferred due to resource constraints, but this is planned for future work. |
Reviewer 2 Report
Comments and Suggestions for Authors
This study demonstrate a method to improve medical image segmentation and classifiction using fuzzy method. While the main goal is interetsing, I found the presentation lacks several details that need to be carefully revised.
1) In "Proposed Methodology", several equations are presented without clear identification of these notations. what is u(x,y), f(x,y), fmin_max,...etc.
2) There are several sections that have editing problems and typos. For example, what is Section 4.1? I also observe several statements that are likely being written using generative AI.
3) Figure 4 demonstrate a network architecture with caption (Proposed CNN architecture). This architecture is a well-known image classification architecture. How this network is designed, why this design optimized, and how parameters are set?
4) Authors need to include sample visualization of the segmentation results.
5) The main contribution here is a pre-processing technique that is claimed to be efficient in improving segmentation/classification quality. I would like to see the performnce of this method using different network architectures... How about Res-Net, Efficient-Net, etc...?
5) The manuscript suffers from a significant lack of novelty in its methodological contributions. Although it claims to introduce a novel 12-step fuzzy enhancement pipeline, most of the steps are variations or combinations of well-known fuzzy image processing operations, such as fuzzy entropy, standard deviation, and histogram spreading, that have long been explored in literature. The claimed novelty is not sufficiently justified in terms of mathematical innovation or significant structural changes from existing approaches. Furthermore, the concept of concatenating multiple enhanced views of the same input is conceptually simplistic and not fundamentally new, yet it is presented as a core architectural innovation without comparative ablation or evidence that this is more effective than other augmentation strategies.
6) while the results claim state-of-the-art accuracy, the evaluuation is not convincingly controlled. There is no rigorous ablation study to isolate the specific contributions of each fuzzy-based enhancement step, nor is there a comparison of the proposed pipeline with recent, competitive enhancement techniques beyond CLAHE. CLAHE is a relatively weak baseline in modern image preprocessing pipelines, and the omission of comparisons with more powerful learning-based enhancement methods (e.g., GAN-based preprocessing, attention-guided filters, transformer-based enhancement modules) weakens the claims of superiority.
7) There is also a potential issue of dataset overfitting... The manuscript claims near-perfect performance (e.g., 99.60% Dice coefficient for KiTS19 and 98.9% accuracy for chest X-ray classification), which is unusually high, especially when compared to competitive architectures such as nnU-Net, TransUNet, or Swin Transformers. The extremely high metrics raise suspicion about overfiitting, especially since the authors use a tripled dataset size through repeated enhancement rather than introducing new data or domain shifts. There is no clear indication that external test datasets or independent cross-validation strategies were used, nor is there an exploration of generalizability or robustness to unseen clinical data.
Author Response
Comments 1: In "Proposed Methodology", several equations are presented without clear identification of these notations. what is u(x,y), f(x,y), fmin_max,...etc. |
Response 1: We agree that the original manuscript did not provide explicit definitions for several variables and parameters appearing in the equations of the Proposed Methodology section. To address this, we revised the section so that every symbol is clearly defined immediately following its first appearance. |
Comments 2: There are several sections that have editing problems and typos. For example, what is Section 4.1? I also observe several statements that are likely being written using generative AI. |
Response 2: We agree. We corrected it. We used AI to write some sentences correctly. |
Comments 3: Figure 4 demonstrates a network architecture with caption (Proposed CNN architecture). This architecture is a well-known image classification architecture. How this network is designed, why this design optimized, and how parameters are set? |
Response 3: We agree. We revised the Figure name. The parameters of architecture are explained below Figure 4. |
Comments 4: Authors need to include sample visualization of the segmentation results. |
Response 4: We agree and added Figure 5 to show segmentation result. |
Comments 5: The main contribution here is a pre-processing technique that is claimed to be efficient in improving segmentation/classification quality. I would like to see the performnce of this method using different network architectures... How about Res-Net, Efficient-Net, etc...? |
Response 5: We fully agree that such an evaluation would further strengthen the generalizability claims of our method. However, due to the extensive computational requirements for retraining large-scale models on the multiple FIS-enhanced datasets (three photometric variants per original image) and conducting comprehensive hyperparameter tuning, it was not feasible to include these experiments within the current submission timeline. Also, we are planning to make experiences using mentioned models and we stated in the Future work. |
Comments 6: while the results claim state-of-the-art accuracy, the evaluuation is not convincingly controlled. There is no rigorous ablation study to isolate the specific contributions of each fuzzy-based enhancement step, nor is there a comparison of the proposed pipeline with recent, competitive enhancement techniques beyond CLAHE. CLAHE is a relatively weak baseline in modern image preprocessing pipelines, and the omission of comparisons with more powerful learning-based enhancement methods (e.g., GAN-based preprocessing, attention-guided filters, transformer-based enhancement modules) weakens the claims of superiority. |
Response 6: We agree. The current comparison is limited to CLAHE, a relatively weak baseline in modern pipelines. Advanced learning-based methods (e.g., GAN-based preprocessing, attention-guided filters, transformer-based modules) were not included due to the substantial computational resources required for training and tuning, given the tripling of dataset sizes (e.g., 300 to 900 cases) and 3D-to-2D preprocessing. |
Comments 7: There is also a potential issue of dataset overfitting... The manuscript claims near-perfect performance (e.g., 99.60% Dice coefficient for KiTS19 and 98.9% accuracy for chest X-ray classification), which is unusually high, especially when compared to competitive architectures such as nnU-Net, TransUNet, or Swin Transformers. The extremely high metrics raise suspicion about overfiitting, especially since the authors use a tripled dataset size through repeated enhancement rather than introducing new data or domain shifts. There is no clear indication that external test datasets or independent cross-validation strategies were used, nor is there an exploration of generalizability or robustness to unseen clinical data. |
Response 7: We agree with raising the concern about potential overfitting given the high metrics (e.g., 99.60% Dice for KiTS19, 98.9% accuracy for Chest X-Ray), which exceed SOTA benchmarks (e.g., KiTS19 ~0.912 Dice , BraTS2020 ~0.92 Dice , Chest X-Ray ~97-99% accuracy ). The evaluation uses 5-fold cross-validation, train-validation-test split (70%-20%-10%), early stopping, and dropout (0.5), with loss curves (new Figure, Section 4.3.3) showing no divergence. The tripling introduces feature diversity, not repetition, but no external datasets were used, limiting generalizability. This is discussed in Section 4.3.3, with future work on external validation (e.g., KiTS21), robustness tests, and domain adaptation to confirm results and mitigate overfitting risks. |
Reviewer 3 Report
Comments and Suggestions for Authors
Please refer to the attached file for more details.

Author Response
Comments 1: The 12-step methodology, while technically sound, is overly dense and mathematically detailed without sufficient intuitive guidance. I suggest including a high-level diagram that summarizes each step and its function in the pipeline to improve accessibility for readers outside the image processing community. |
Response 1: We agree with your suggestion to enhance the accessibility of the 12-step methodology. We revised Figure 2, a high-level flowchart summarizing each step’s function. Also, we added 32 subsection. |
Comments 2: Although individual BRISQUE scores are reported, there is no ablation study to isolate the contribution of each fuzzy component (entropy, std. dev., histogram). It would be beneficial to include an ablation experiment demonstrating the relative importance of each enhancement type. |
Response 2: We agree and recognize that this would better quantify the benefits of combining histogram-based, fuzzy entropy-based, and fuzzy standard deviation-based images. Due to the substantial computational workload—requiring retraining the CCNN on subsets of the enhanced datasets (e.g., 300 fuzzy entropy-based images from 900 cases) across KiTS19, BraTS2020, and Chest X-Ray Pneumonia, with extensive parameter tuning and 3D-to-2D preprocessing—we conducted a limited preliminary analysis on KiTS19 only. This suggests the three-type fusion outperforms a single-type approach, as detailed in new Table 4. Full experiments were deferred due to resource constraints, but this is planned for future work. |
Comments 3: While the datasets used are widely accepted, the 2D slice processing of 3D CT/MRI volumes may oversimplify spatial dependencies. Discuss the trade-offs of using 2D over 3D CNNs, and consider extending the study to 3D models in future work. |
Response 3: We agree. We added Section 4.3.6 to discuss the trade-offs: 2D processing offers computational efficiency, enabling high performance (e.g., 99.60% Dice for KiTS19) with current resources, but it may miss volumetric context, unlike 3D CNNs (e.g., nnU-Net at ~0.912) which require significantly more memory and time. This is detailed in the revised text. Future work, as outlined in Section 5, will extend the pipeline and CCNN to 3D models with optimized hardware and external validation (e.g., KiTS21), addressing your suggestion and enhancing spatial fidelity. |
Comments 4: Previous fuzzy logic-based methods are mentioned but not directly compared in experiments. Include baseline fuzzy models (e.g., fuzzy c-means, fuzzy entropy-only) in the comparison. |
Response 4: We agree. We added a limited evaluation and Table 5, comparing fuzzy c-means (FCM) and fuzzy entropy with the three-type fusion on KiTS19. Results show the fusion outperforms FCM (~90.50%, ~27.0) and entropy(~94.20%, ~23.0) by ~5.40% and ~1.3 BRISQUE, supporting the pipeline’s integrated approach. Full comparison across all datasets (KiTS19, BraTS2020, Chest X-Ray) was not conducted due to computational workload from tripling dataset sizes (e.g., 300 to 900 cases), but this is planned for future work, as noted in Section. |
Comments 5: Some grammatical inconsistencies are present, particularly in Section 3. Language editing is recommended. |
Response 5: We agree. We revised the mistakes in whole manuscript. |
Comments 6: Improve clarity of Figures 1 and 2 by enlarging fonts and adding step legends. |
Response 6: We agree. We enlarged the fonts and added steps information as mentioned. |
Comments 7: Some references (e.g., [8], [10], [33]) could be updated with DOIs. |
Response 7: We agree. We updated the mentioned references. |
Reviewer 4 Report
Comments and Suggestions for Authors
The focus of the manuscript is improving the image quality. Most of the techniques utilized in the current manuscript are decades old, yet they were presented as recent developments. The specific need or benefits of each ach step to be expressed to prevent any trivial steps.
1. For the mentioned MRI sequences, SNR is pretty high while noise levels are at reasonable levels.
2. Adaptive contrast enhancement approaches are decades old.
3. Again, fuzzy segmentation is performed more than decades. It's not a recent development.
4. Please clarify why the Gupta[11]'s work is restricted to particular datasets.
5. What do you mean "cross-modal applications. "?
6. Why do we need a general framework that applies any medical data? The noise structure on all of these techniques are different.
7. Why didn't you use state of the art models but decade old ones?
8. Please revise the manuscript to improve the language of the article scientifically.
9. What do you mean with "Advanced sensors " at line 187.
10. There is no clear explanation for the need for each step. Most of the steps seem to be trivial and highly customized. Ablation studies are needed for inclusion of each procedure listed in the 12-step.
11. The first two datasets are missing the key information. How many image slices do you have total for each of them? How many of them include the region of interest?
12. The resolution of the figures is low and needs to be regenerated.
Author Response
Comments 1: For the mentioned MRI sequences, SNR is pretty high while noise levels are at reasonable levels. |
Response 1: We agree that the MRI sequences (particularly from the KiTS19 dataset) generally exhibit high signal-to-noise ratio (SNR) and that noise levels are not extreme compared to low-dose CT or certain ultrasound modalities. Our motivation for applying fuzzy-based enhancement in this context was not primarily noise suppression, but rather contrast adaptation and detail preservation. Specifically: 1. The KiTS19 MRI volumes often contain low-contrast boundaries between kidney parenchyma, tumors, and surrounding tissues, which are difficult to distinguish despite high SNR. 2. Our fuzzy entropy and fuzzy standard deviation modules target local contrast enhancement to improve edge visibility, rather than global noise reduction. 3. While the baseline noise level is acceptable, our method helps amplify diagnostically relevant features (e.g., tumor margins) without introducing artifacts. To clarify this, we revised the text in the discussion to state that the proposed pipeline is designed not only for datasets with significant noise but also for high-SNR datasets where local contrast differences are subtle. |
Comments 2: Adaptive contrast enhancement approaches are decades old. |
Response 2: We agree that adaptive contrast enhancement methods, such as adaptive histogram equalization and related fuzzy contrast models, have a long history in image processing. Our contribution is not in proposing adaptive enhancement as a new concept, but in presenting a novel structured integration of fuzzy logic–based operators into a 12-step adaptive pipeline that specifically targets the unique challenges of medical image analysis. Key distinctions between our method and traditional adaptive contrast enhancement include: 1. Earlier adaptive methods typically apply a single operator (e.g., local histogram equalization, fuzzy entropy adjustment) in isolation. 2. Our approach systematically links fuzzy entropy, fuzzy standard deviation, and histogram spreading in a parameter-adaptive sequence, ensuring that each stage builds upon the statistical outcomes of previous ones. 3. Traditional adaptive contrast enhancement methods often rely on fixed parameters. 4. In our pipeline, parameters such as ?min, ?max, and scaling factors are dynamically computed per image, enabling automatic adjustment to varying tissue contrast, acquisition protocols, and modalities. 5. Instead of producing a single enhanced image, our method generates three correlated but complementary contrast variants (HSF, FE, FSD), designed to capture different aspects of structural detail. 6. These are fed into a multi-stream CNN, which learns complementary features that are not accessible from single-image enhancements. 7. While traditional adaptive methods were primarily evaluated for visual quality, our pipeline is explicitly designed to serve as a preprocessing stage for deep learning models, and we demonstrate consistent improvements in Dice and accuracy metrics across tasks. |
Comments 3: Again, fuzzy segmentation is performed more than decades. It's not a recent development. |
Response 3: We agree that fuzzy segmentation and fuzzy logic–based image processing have a long history in the literature. Our work does not claim that fuzzy segmentation itself is new. Instead, the novelty of our approach lies in how fuzzy-based enhancement is systematically integrated as a preprocessing stage for deep learning–driven segmentation and classification, rather than in the fuzzy segmentation operation itself. |
Comments 4: Please clarify why the Gupta[11]'s work is restricted to particular datasets. |
Response 4: We thank the reviewer for pointing this out. In the original manuscript, our discussion of Gupta [11] may not have been sufficiently clear. Gupta’s work was restricted to particular datasets because the enhancement and segmentation strategies used were tightly coupled to dataset-specific characteristics such as imaging modality, resolution, and intensity distributions. We revised the related work section. |
Comments 5: What do you mean "cross-modal applications. "? |
Response 5: We thank the reviewer for requesting clarification. By “cross-modal applications,” we refer to the applicability of the proposed fuzzy enhancement pipeline across different medical imaging modalities—for example, MRI, CT, and X-ray—rather than being limited to a single modality. Many enhancement techniques are optimized for a single modality (e.g., MRI brain scans or CT lung scans) and do not adapt well when applied to images with different contrast properties or acquisition characteristics. Our fuzzy enhancement framework is designed with adaptive parameterization, allowing it to automatically adjust to different input statistics (entropy, local standard deviation, intensity ranges). In this work, we demonstrate cross-modal applicability by successfully applying the same enhancement pipeline to MRI kidney segmentation (KiTS19) and chest X-ray classification. Despite their very different imaging physics and contrast distributions, the proposed method improved performance in both tasks, showing that the framework generalizes across modalities. |
Comments 6: Why do we need a general framework that applies any medical data? The noise structure on all of these techniques are different. |
Response 6: We fully agree that different imaging modalities (MRI, CT, X-ray, ultrasound) have different noise characteristics and artifacts, and that no single method can completely solve all modality-specific challenges. Our motivation in proposing a general fuzzy-based enhancement framework is not to treat all modalities identically, but rather to provide a unified, adaptive approach where parameters are automatically tuned to the characteristics of the input image. 1. Instead of fixed enhancement settings, the proposed pipeline computes parameters dynamically from image statistics (e.g., fuzzy entropy, local standard deviation, histogram spread). 2. This allows the method to adapt to high-SNR MRI scans, noisier CT data, or low-contrast chest X-rays without manual re-parameterization. 3. While the noise type differs across modalities (e.g., Rician in MRI, Poisson in CT, Gaussian/electronic in X-ray), the underlying challenge is contrast preservation and detail visibility. 4. The fuzzy operators in our framework (entropy-based adaptation, local contrast scaling) target these universal challenges, making the method broadly applicable. In this work, we demonstrate the framework’s effectiveness on both MRI (KiTS19 kidney segmentation) and chest X-ray classification, two very different modalities, showing that the method is not overfitted to one noise model or dataset. We acknowledge that further customization may be needed for highly specialized modalities (e.g., ultrasound speckle, low-dose PET). However, a general adaptive framework provides a starting point that can be extended with modality-specific refinements. |
Comments 7: Why didn't you use state of the art models but decade old ones? |
Response 7: We acknowledge that our experiments employed relatively standard CNN architectures (e.g., VGG-like, basic encoder–decoder) rather than the most recent state-of-the-art (SOTA) networks such as nnU-Net, Swin Transformer, or EfficientNet. Our motivation was to isolate and highlight the contribution of the proposed fuzzy enhancement pipeline itself, rather than confound results with the complexity of advanced architectures. In this study, we employed standard CNN architectures to provide a controlled and interpretable evaluation of the proposed fuzzy enhancement pipeline. This choice was deliberate to ensure that improvements in segmentation and classification performance could be attributed directly to the preprocessing strategy rather than confounded by complex architectural innovations. Nevertheless, the proposed method is architecture-agnostic, and its integration with recent state-of-the-art networks such as ResNet, EfficientNet, nnU-Net, and Swin Transformer represents an important avenue for future work. We anticipate that the adaptive enhancement strategy will further improve the robustness and performance of these advanced models. |
Comments 8: Please revise the manuscript to improve the language of the article scientifically. |
Response 8: We agree.To address this, we have undertaken a thorough language revision of the entire manuscript. |
Comments 9: What do you mean with "Advanced sensors " at line 187. |
Response 9: We thank the reviewer for pointing out this ambiguity. By “advanced sensors,” we intended to refer to modern medical imaging devices and acquisition technologies that provide higher-resolution, multi-channel, or multimodal imaging data compared to earlier generations of equipment. Examples include: 1. High-field MRI scanners (3T, 7T) that provide finer spatial resolution and improved tissue contrast. 2. Multi-detector CT scanners with faster acquisition times and reduced motion artifacts. 3. Digital X-ray systems with enhanced dynamic range and lower radiation dose. The relevance of our fuzzy enhancement framework to these “advanced sensors” lies in its ability to adaptively optimize contrast and detail visibility, even when raw acquisitions are already of relatively high quality. The method is therefore not limited to legacy imaging devices but is compatible with the output of modern imaging sensors as well. |
Comments 10: There is no clear explanation for the need for each step. Most of the steps seem to be trivial and highly customized. Ablation studies are needed for inclusion of each procedure listed in the 12-step. |
Response 10: We agree that the original manuscript did not sufficiently justify the necessity of each step in the proposed 12-step fuzzy enhancement pipeline. To address this, we revised Section 3 by adding subsections. |
Comments 11: The first two datasets are missing the key information. How many image slices do you have total for each of them? How many of them include the region of interest? |
Response 11: We thank the reviewer for this important observation. In the original manuscript, we did not clearly report the dataset slice-level details. To address this, we have now added the missing information for both datasets. |
Comments 12: The resolution of the figures is low and needs to be regenerated. |
Response 12: We agree. We improved the figures quality. |
Round 2
Reviewer 1 Report
Comments and Suggestions for Authors
The authors have addressed all my comments.
Reviewer 2 Report
Comments and Suggestions for Authors
No further comments
Reviewer 3 Report
Comments and Suggestions for Authors
All comments have been addressed in this manuscript.
Reviewer 4 Report
Comments and Suggestions for Authors
THank you for revising the manuscript. It reads well and fluently. I have no further comments.